# Pre-meiotic H1.1 degradation is essential for Arabidopsis gametogenesis

Yanru Li[1,5], Danli Fei[1,5], Jasmin Schubert[1,5], Kinga Rutowicz[1,3], Zuzanna Kaczmarska[2,4], Alberto Linares[1], Alejandro Giraldo Fonseca[1], Sylvain Bischof[1], Ueli Grossniklaus [1] & Célia Baroux [1]✉

## Abstract

**Despite being evolutionary distant, plants and animals exhibit a shared phenomenon during the transition from somatic-to-reproductive cell fate marked by extensive structural and compositional changes in chromatin. This chromatin reprogramming occurs in the plant SMCs (Spore Mother Cells) and animal PGCs (primordial germ cells) and is initiated by the loss of linker histones (H1). H1 loss is essential to establish pluripotency in animal PGCs but its role is not known in plants. Here, we identified two regulatory pathways involving a citrullinase and an E3-ubiquitin ligase that contribute H1.1 loss in female SMCs in Arabidopsis. We also identified roles for two specific residues: an arginine, whose positive charge contributes to H1.1 destabilization from chromatin, and a lysine in the globular domain that is essential for H1.1 degradation. Ovules with impaired H1.1 loss in the SMC proceed through sporogenesis but fail to complete gametogenesis. We propose that a citrullination–ubiquitination pathway governs pre-meiotic H1 depletion as a critical mechanism for establishing post-meiotic competence in the Arabidopsis germline.**

**Keywords** Linker Histone; Citrullination; Ubiquitination; Gametogenesis; *Arabidopsis*
**Subject Categories** Chromatin, Transcription & Genomics; Plant Biology

## Introduction

Sexually reproducing multicellular organisms form a specialized cell lineage dedicated to meiosis and gametogenesis, two processes that are fundamental to ensuring genetic diversity while maintaining chromosomal integrity across generations. In mammals, this lineage arises from primordial germ cells (PGCs), which are specified early in development and set aside from the pluripotent embryonic cell mass. Following meiosis, these cells directly differentiate into specialized gametes (Bendel–Stenzel et al, 1998). In flowering plants, the functional equivalent of PGCs are spore mother cells (SMCs), which form later in development within specific floral tissues

of the adult organism. In contrast to animal meiotic products, the haploid spores of plants are pluripotent and give rise to a multicellular gametophyte, containing several distinct cell types, including the gametes (Skinner and Sundaresan, 2018). This difference in timing, origin, and developmental competence of the meiotic products raises questions about the mechanisms that regulate the de novo specification of SMCs in plants and the establishment of a pluripotent state following meiosis.

Recent studies have uncovered multiple layers of regulation controlling the establishment of female SMC in the model plant *Arabidopsis*. Their specification involves geometrical information, hormonal cues, cell–cell signaling modules, genetic and epigenetic factors (Pinto et al, 2019; Lora et al, 2019; Jiang and Zheng, 2022; Hernandez-Lagana et al, 2021; Cai et al, 2022, 2025; Huang et al, 2023). Typically, female SMCs are initiated by the enlargement of the most distal cell in the subepidermal layer (L2) of ovule primordia (Pinto et al, 2019; Lora et al, 2019). SMC differentiation, defined by distinct morphological features and markers, is a gradual process tightly linked to the growth of the ovule primordium (Hernandez-Lagana et al, 2021). While the mechanisms are not fully elucidated, several non-cell autonomous mechanisms have been identified that govern both SMC formation and singleness. SMC specification involves a transcriptional co-repressor complex, consisting of SPOROCYTELESS/NOZZLE (SPL/NZZ) (Balasubramanian and Schneitz, 2000) and TOPLESS/TOPLESS-RELATED (TPL/TPR) proteins (Wei et al, 2015). This complex likely acts indirectly to enable the expression of *WUSCHEL* (WUS), a homeodomain transcription factor. WUS subsequently activates the expression of *WINDHOSE1* (*WIH1*) and *WIH2* at the apex of the primordium that, together with the tetraspanin-type protein TETRASPANIN1/TORNADO2/EKEKO (TET1/TRN2), contribute to initiate SMC specification (Lieber et al, 2011; Pinto et al, 2019; Lora et al, 2019). Furthermore, the singleness of SMCs is controlled by lateral inhibition involving small interfering RNAs (siRNAs) and hormones, i.e., auxin and brassinosteroids, that prevent neighboring cells from adopting an SMC fate (Cai et al, 2022; Lora et al, 2019; Grossniklaus and Schneitz, 1998; Cai et al, 2025). Growth of the primordium itself also contributes to SMC singleness in which the shape of its apex constrains the domain of action of key regulators together with mechanical cues supporting SMC expansion (Hernandez-Lagana et al, 2021). SMC singleness is further secured by cell cycle regulators, including KIP-RELATED

[1]Department of Plant and Microbial Biology & Zurich-Basel Plant Science Center, University of Zurich, Zurich, Switzerland. [2]European Molecular Biology Laboratory (EMBL) Grenoble, Grenoble, France. [3]Present address: Institute of Molecular Plant Biology, ETH Zurich, Zurich, Switzerland. [4]Present address: Laboratory of Protein Structure, International Institute of Molecular and Cell Biology, Warsaw, Poland. [5]These authors contributed equally: Yanru Li, Danli Fei, Jasmin Schubert. ✉E-mail: cbaroux@botinst.uzh.ch

PROTEINS (KRPs), also known as INHIBITORS OF CYCLIN-DEPENDENT KINASES (ICKs), and CYCLIN-DEPENDENT KINASES (CDKs), which help prevent mitotic division of the SMC, by inhibiting RETINOBLASTOMA-RELATED1 (RBR1), an inhibitor of WUS (Zhao et al, 2017; Cao et al, 2018). Thus, SMC specification results from the spatio-temporal integration of an array of intrinsic and extrinsic signals.

Ultimately, SMCs are committed to undergo meiosis, producing functional haploid spores that give rise to gametophytes that harbor the gametes. The SMC transcriptome exhibits signatures of reprogramming consistent with cell fate change (Schmidt et al, 2011; Hou et al, 2021) but the underlying epigenome has largely remained inaccessible to profiling studies. Yet, the SMC chromatin undergoes drastic structural and compositional changes compared to that of neighboring cells. These changes include the loss of linker histone variants H1.1 and H1.2 and of nucleosomal histone variants H3.1/HTR13 and H2AZ/HTA11, elevated levels of H3K4me3 and H4ac, reduced levels of H3K27me3 and DNA methylation in the CHH context (Ingouff et al, 2017; She et al, 2013). Interestingly, however, the overall transcriptional activity remains low in SMCs compared to neighboring cells (She et al, 2013).

The loss of H1 linker histones in Arabidopsis SMCs (She et al, 2013) is reminiscent of the loss of somatic H1 subtypes (H1.1–H1.5 and H1.10) in mouse PGCs during early pre-implantation stage (E11.5) (Hajkova et al, 2008; Izzo et al, 2017). H1 variants were long considered to serve only a structural role in chromatin folding, hindering transcription. But recent studies uncovered a more complex interaction between H1 histones and epigenetic regulation, including DNA methylation, histone methylation and acetylation, both in plants and animals (reviewed in Fyodorov et al, 2018; Wierzbicki and Jerzmanowski, 2005; Zemach et al, 2013; Rutowicz et al, 2019; Choi et al, 2020; He et al, 2024; Teano et al, 2023). This raises the question whether the loss of H1 in plant SMCs and mouse PGCs controls cellular reprogramming associated with the somatic-to-reproductive transition. Supporting this hypothesis, H1 depletion in mouse PGC was found critical for establishing pluripotency within the germline (Christophorou et al, 2014). Similarly, H1 depletion in mouse embryonic stem cells leads to the upregulation of pluripotency genes, causing the cells to stall in a self-renewal state with impaired differentiation potential (Zhang et al, 2012).

In plants, the role and mechanisms of H1 loss in Arabidopsis SMCs are unknown. Here we report on two pathways involving the citrullinase AGMATINE IMINO HYDROLASE (AIH) and the E3 ubiquitin ligase CULLIN4 that regulate H1.1 loss in the SMC, involving two key residues of H1.1, an arginine R57 and lysin K89. We propose a working model involving arginine citrullination as a mechanism increasing H1.1 mobility and ubiquitination leading to degradation. Importantly, we found that disrupting H1 loss in the SMC does not affect meiosis but instead compromises gametogenesis suggesting that pre-meiotic chromatin dynamics contributes to post-meiotic fate.

## Results

### The E3-ubiquitin ligase CULLIN4 contributes to H1.1 degradation in the SMC

In previous work, we reported the loss of H1.1 and H1.2 in *Arabidopsis* male and female SMCs but not in neighboring cells (She and Baroux, 2015; She et al, 2013). As this process was sensitive to the proteasome inhibitor Syringolin A (She et al, 2013) we hypothesized a ubiquitin-mediated targeting of H1 to the proteasome, a major pathway for protein

degradation in plants (Vierstra, 1993). E3 ligases of the CULLIN family which transfer ubiquitin to the target protein (Yang et al, 2021) were thus prime candidates. Published RNA in situ hybridization indicated *CULLIN4 (CUL4)* expression in young ovule primordia (Chen et al, 2006) and microarray data confirmed expression in female SMCs and surrounding nucellus (Fig. EV1A). Thus, if CULLIN4 were involved in H1 degradation, its downregulation at the onset of ovule primordium development should prevent the loss of H1 in mature SMCs. CUL4 is critical to plant development, and its downregulation leads to pleiotropic defects in leaves and roots as well as severe sterility (Chen et al, 2006; Bernhardt et al, 2006). Complete loss of function is embryo-lethal and is also associated with a mild reduction (10–15%) in transmission efficiency through both male and female gametes, indicating a low level of gametophytic lethality (Dumbliauskas et al, 2011). To avoid confounding effects, we conditionally knocked down *CUL4* expression in developing ovule primordia just prior to the onset of H1 depletion. For this, we expressed an artificial microRNA targeting *CUL4, amiR[CUL4]* (Fig. EV1B) under control of the *pOP/LhGR* Dexamethasone (Dex)-inducible system (Craft et al, 2005; Samalova et al, 2019). We confirmed a reduction of CUL4 protein levels in seedlings induced for *amiR[CUL4]* expression and observed the known *fusca* phenotype typical of a *cul4* loss-of-function mutation (Chen et al, 2006) (Fig. EV1C,D). To investigate the possible role of CUL4 in H1 depletion during SMC development, we introgressed the inducible *amiR[CUL4]* line into an H1.1-GFP reporter line (She et al, 2013). We rationalized that if CUL4 played a role in this process, knock-down of its expression should lead to partial or full retention of H1.1-GFP in the SMC. *amiR[CUL4]* expression was induced in young flower buds in planta as previously described (Schubert et al, 2022). Five days post induction (5 dpi), flower buds were collected, and ovule primordia at stage 2-I/2-II were scored under confocal microscopy for H1.1-GFP signal considering two categories: primordia showing no detectable signal in the SMC ("H1.1 depletion") and those showing a detectable signal in the SMC ("H1.1 persistence"). The Mock control showed 11% ($n = 137$) primordia with a residual H1.1-GFP signal, likely due to the gradual nature of H1.1 loss. By contrast, Dex-treated flower buds harbored 52% ($n = 178$) primordia with a strong H1.1-GFP signal in the SMC (Fig. 1A). We obtained similar results when primordia were treated with the proteasome inhibitor epoxomicin (Fig. 1B). We concluded that *amiR[CUL4]* expression in young ovule primordia can impair H1.1 depletion during SMC formation, likely due to effective *CUL4* downregulation. The delicate experimental setup, involving manual treatment of young flower buds for induction and the use of an artificial miRNA might explain why the H1.1 persistence phenotype was not fully penetrant. But this approach has the benefit of allowing spatial and temporal control of *CUL4* downregulation, minimizing confounding effects caused by constitutive loss-of-function. Collectively, our results indicate that the E3-ubiquitin ligase CULLIN4 plays a role in H1.1 degradation during SMC development.

### A screen for potential ubiquitination targets identifies K89 in the globular domain of H1.1

E3-ligases preferentially target lysine-rich regions (Yang et al, 2021). Given that linker histones are particularly rich in lysine throughout their protein sequence (Jerzmanowski, 2004), we considered that ubiquitination of one or several lysines could drive H1 degradation. To test this hypothesis, we first created six H1.1 mutant variants with batches of K-to-R amino acid substitutions (Fig. 1C). Candidate lysines were selected based on computational

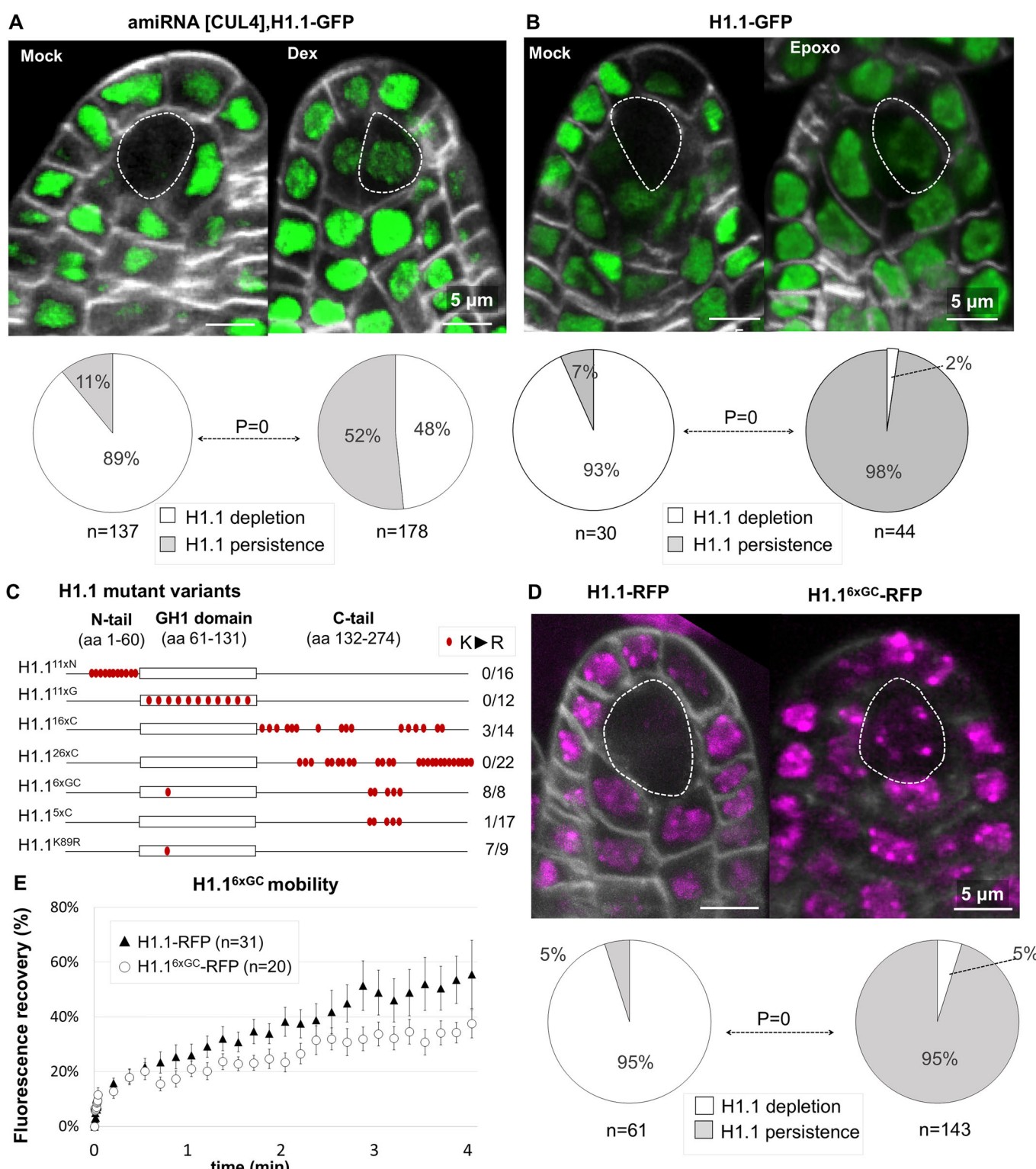

predictions (AtH1.1 K134, 139, 144, 172, 177, 179, 185, 189, 191, 193, 204, 211, 213, 215, 223, 226, 232, 273, 274), (Chen et al, 2011; Radivojac et al, 2010; Walton et al, 2016), based on proteomic evidence of ubiquitination in seedling tissues (AtH1.1 K89, 204, 206, 211, 213, 215, Fig. EV1E), (Walton et al, 2016) and by comparing with ubiquitinated sites in mouse and human counterparts (AtH1.1 K77, 89, 104, 111), (Wiśniewski et al, 2007). We also included in this list of candidate residues, several lysins in the N-terminal tail upstream the globular domain to cover all three functional domains of the protein in our targeted mutagenesis

**Figure 1.  H1.1 degradation is mediated by the E3-Ubiquitin ligase CULLIN 4.**

(A, B) H1.1-GFP degradation in the SMC is impaired when inducing the expression of an artificial miRNA against *CUL4* (**A**, *amiRNA[CUL4]*) or when treating with the proteasome inhibitor epoxomicin (**B**). Young inflorescences were treated either with 10 µM dexamethasone or a mock solution as control (**A**, Dex, Mock) (Craft et al, 2005; Samalova et al, 2019), or with a 5 µM epoxomicin solution or Mock solution as control (**B**, Epoxo, Mock). Images show partial projections of confocal image series showing the GFP signal (green) and cell wall stained with Renaissance (Musielak et al, 2015) (gray). Dotted borders outline the Spore Mother Cell (SMC). Pie charts below the images show the proportion of ovule primordia at stage 2-I/2-II five days post induction (5 dpi) showing either H1.1-GFP depletion (white) or persistence (gray) in the SMC. *n*, number of primordia scored. *P* value, Fischer-exact test (α < 0.05). (**C**) Schematic representation of H1.1 mutant variants carrying several K-to-R amino acid substitutions (red dots) in their N-terminal tail, globular domain, C-terminal tail, or a combination thereof, and fused to RFP. The numbers on the right indicate the number of independent lines showing a persistent signal in the SMC (indicating a lack of degradation) over (/) the total number of lines screened. (**D**) Representative images of H1.1-RFP and H1.1$^{6xGC}$RFP expression in ovule primordia at stage 2-I, 5 dpi. Dotted borders outline the Spore Mother Cell (SMC). Pie charts show the proportion of ovule primordia showing either H1.1-RFP depletion (white) or persistence (gray) in the SMC. *n*, number of primordia scored. *P* value, Fischer-exact test (α < 0.05). (**E**) Fluorescence recovery after photobleaching (FRAP) of H1.1-RFP and H1.1$^{6xGC}$RFP in root nuclei from 5-day-old seedlings grown in the presence of Dex. *n*: number of analyzed nuclei; error bar: standard error to mean. See also Source data Fig. 1, Table EV1 and replicate experiments in Fig. EV1 and Source Data EV1. Source data are available online for this figure.

approach. Collectively, six mutant H1.1 variants were designed to carry K-to-R mutations in the N-terminal domain, the globular domain, the C-terminal tail or a combination thereof. The mutant variants were expressed as RFP translational fusions under the control of the Dex-inducible *pOP/LhGR* system as before (Craft et al, 2005; Samalova et al, 2019). We screened primary transformants following induction of young flower buds, for H1.1-RFP depletion *versus* persistence in the SMC of ovule primordia at stage 2-I/2-II, under confocal microscopy. None of the 16 and of the 12 transformants expressing mutant variants with 11 K-to-R substitutions in the N-terminal tail (H1.1$^{11xN}$) or in the globular domain (H1.1$^{11xG}$), respectively, showed persistence, neither did any of the 26 transformants expressing a mutant variant with 26 substitutions in the C-terminal tail (H1.1$^{26xC}$) (Fig. 1C). In 3 lines expressing a variant with 16 substitutions in the C-terminal tail (H1.1$^{16xC}$) we observed some residual H1.1$^{16xC}$-RFP signal, but since it was not reproduced in the other 11 independent transformants we did not consider this mutant variant for further analyses (Fig. 1C). However, all the eight transformants expressing H1.1$^{6xGC}$, a variant with 6 substitutions (five in the C-terminal tail, one in the globular domain) showed a strong persistence of H1.1$^{6xGC}$-RFP in the SMC (Fig. 1C,D). This H1.1 variant was mutated in all six sites shown to be ubiquitinated in seedlings (Walton et al, 2016, Fig. EV1E). Typically, 76–95% of the ovule primordia expressing H1.1$^{6xGC}$-RFP showed a clear signal in the SMC at stage 2-I/2-II (Fig. 1D and Fig. EV1F for replicates). In contrast, primordia expressing a native H1.1 variant fused to RFP (H1.1-RFP) under control of the same Dex-inducible system, showed depletion in 95% of the SMCs in ovule primordia at stage 2-I/2-II (Fig. 1D), with the remaining 5% showing only weak, residual signals. The persistence of H1.1$^{6xGC}$-RFP could be due to either a lack of ubiquitination and thus degradation, or enhanced stability on chromatin. Because profiling H1.1 post-translational modifications (PTMs) in Arabidopsis SMCs is currently out of reach, we assessed H1.1$^{6xGC}$ stability by Fluorescence Recovery After Photobleaching (FRAP). FRAP was previously used to investigate turnover and residency time of H1 mutant variants in vivo, in order to identify residues involved nucleosome binding (Brown et al, 2006). We compared the fluorescence recovery rates of H1.1-RFP and H1.1$^{6xGC}$-RFP following photobleaching in Dex-treated seedling root nuclei. The data indicated a similar mobility, both during the initial, rapid recovery phase and later, during the slow recovery phase (Fig. 1E and Fig. EV1G). This suggests that substitution of the six lysines into arginines—harboring a similar charge—did not

enhance H1.1 stability. Considering that these lysines are ubiquitinated in seedling tissues (Walton et al, 2016), our findings strongly suggest that ubiquitination, and consequently, degradation of H1.1$^{6xGC}$-RFP is compromised.

To further distinguish the contribution of the different lysines mutated in the H1.1$^{6xGC}$ variant, we created two additional mutants: one carrying a single mutation at lysine 89 (K89R) in the globular domain (H1.1$^{K89R}$, Figs. 1C and EV1H,I) and another with K-to-R mutations at the five remaining lysines in the C-terminal tail (H1.1$^{5xC}$, Fig. 1C). We expressed these variants under the control of the Dex-inducible system and screened several independent transformants. The persistence phenotype was recapitulated in 7 of 9 lines expressing H1.1$^{K89R}$-RFP but in none of 16 lines expressing H1.1$^{5xC}$-RFP indicating that a single amino acid change is sufficient to render H1.1$^{K89R}$-RFP resistant to degradation. Intriguingly, the H1.1$^{11xG}$ variant, which also contains the K89R substitution, did not exhibit the persistence phenotype observed with H1.1$^{K89R}$. Sequencing of the transgenic insertions confirmed the presence of the 11 mutations in 12 lines (Fig. EV1K). The generally lower signals obtained for this mutant variant (Fig. EV1K) suggest lower viability due to cumulated mutations, which may mask the specific effect of K89R.

K89 is exposed to the surface of the third alpha-helix of the globular domain (Fig. EV1H); hence, is unlikely to contribute to interactions with the nucleosome. In line with this idea, mutation of K52 in the mouse MmH1(o) variant, homologous to K89 in AtH1.1 (Fig. EV1H) did not change H1(o) mobility (Brown et al, 2006). Similarly, the K89R mutation in the *Arabidopsis* variant did not affect the overall mobility of H1.1 (Fig. EV1G). Finally, we observed that both H1.1$^{6xGC}$-RFP and H1.1$^{K89R}$-RFP showed a nuclear distribution like the native H1.1-RFP, present in both euchromatin and heterochromatin (Fig. EV1J).

In conclusion, we identified a conserved lysine residue, K89, in the globular domain of H1.1 that plays a key role in the degradation of H1.1 in the SMC. The fact that this lysine does not affect binding affinity and was found to be ubiquitinated in seedlings strongly supports a model in which ubiquitination at K89, likely mediated by CUL4, directly or indirectly, drives H1.1 degradation.

## A single arginine residue in the N-terminal tail controls H1 depletion in the SMC

Ubiquitin-mediated protein degradation is a basic cellular process and CUL4 is expressed throughout the ovule primordium

(Dumbliauskas et al, 2011). Hence, we wondered whether another PTM of H1.1 might prime H1.1 for ubiquitination specifically in the SMC. Interestingly, H1.2 depletion in mouse PGCs (Hajkova et al, 2008) is controlled by citrullination of R54, a PTM converting arginine into citrulline (Christophorou et al, 2014). We thus wondered whether a similar citrullination mechanism might operate in the control of H1.1 depletion in *Arabidopsis* SMCs. To investigate this, we designed an H1.1 mutant with an R-to-K substitution preserving the positive charge, at the residue R79 located inside the globular domain, and which is conserved with the mouse homolog R54 residue (Fig. EV2A). We co-expressed H1.1$^{R79K}$-RFP under the control of the *pOP/LhGR Dex*-inducible system and H1.1-GFP as a control (native H1.1 variant expressed under its own promoter, She et al, 2013). H1.1$^{R79K}$-RFP showed depletion in the SMC as did H1.1-GFP (*n* = 260 ovule primordia, 6 independent lines, Fig. EV2B). Then, we mutated another conserved arginine located just before the globular domain, at position R57 (Fig. 2A). This time, the R57K mutation compromised H1.1 depletion in a significant fraction of ovule primordia (26%, *n* = 218 Fig. 2B and 26%, *n* = 209 Fig. EV2C) compared to the control (7% and 3%, Figs. 2B and EV2C) yet not as efficiently as the previously assessed 6xGC and K89R mutations. This suggested that these residues may act differently on H1.1 degradation. In contrast, expressing an H1.1$^{R57A}$-RFP variant, where the R to A substitution results in a neutral charge, showed a residual signal in only 7% SMC with (*n* = 171) similar to the H1.1-RFP controls (Fig. EV2C). This finding indicates that R57 is unlikely the target of a biochemical modification (for instance methylation) but rather suggests a role for its ionic charge. In addition, we observed that the R57K mutation also compromised depletion in male SMCs, whereas the R57A mutation did not (Fig. EV2D).

To assess whether persistence of H1.1$^{R57K}$-RFP in the SMC is due to enhanced chromatin stability compared to H1.1$^{R57A}$-RFP or H1.1-RFP, we conducted FRAP analyses as previously. Strikingly, the R57K mutation did not affect H1.1 mobility, whereas the R57A mutation significantly increased it, as evidenced by a two-fold increase in the recovery rate (Figs. 2C and EV2E). The stability of linker histones binding to the nucleosome is primarily governed by electrostatic interactions with positively charged amino acids on H1 (Brown et al, 2006; Martinsen et al, 2022). Hence, the most likely explanation for the reduced residency time—and faster recovery rate—of H1.1$^{R57A}$ is that the alanine substitution causes charge neutralization and weaker electrostatic interactions between H1.1 and the nucleosome. In support of this hypothesis, we observed a more efficient depletion of H1.1$^{R57A}$ in the SMC of ovule primordia after we refined the analysis by scoring primordia showing a faint signal in heterochromatin ("partial depletion", Fig. EV2A) which we interpret as an intermediate step to full depletion. We noticed that this fraction was significantly reduced in lines expressing H1.1$^{R57A}$-RFP with a concomitant increase of SMC without signal (Fig. EV2A), suggesting a more efficient depletion of H1.1$^{R57A}$ than H1.1$^{R57K}$ or H1.1.

## AGMATINASE IMINOHYDROLYASE, a candidate arginine deiminase is necessary for H1.1 depletion

Following up on our hypothesis that charge neutralization at R57 impacts on H1.1 stability we investigated citrullination as a candidate mechanism. Citrullination, driven by peptidyl arginine deiminases such as the PADI enzymes in animal cells, results in the net loss of a positive charge and has previously been shown to play a role in H1.2 depletion in mouse PGCs (Christophorou et al, 2014). The *Arabidopsis* genome does not encode proteins homologous to PADI enzymes. However, a 3D protein structure homology search identified AGMATINASE IMINOHYDROLYASE (AIH) as closely related to the catalytic domain of PADI4, with a striking conservation of the four catalytic amino acids (Fig. 3A). AIH exhibits citrullinase activity in vitro (Marondedze et al, 2021) and is expressed in young flower buds at stage 8-10 according to publicly available transcriptome datasets (Klepikova et al, 2016) (Fig. EV3A). Using RNA in situ hybridization, we confirmed that *AIH* is expressed in young ovule primordia (Figs. 3B and EV3B). To test whether AIH plays a role in H1.1 depletion we induced the expression of an artificial miRNA against *AIH* (*amiR[AIH]*, Fig. EV3C) using the Dex-induction system, in an H1.1-GFP expressing line. Consistent with our hypothesis, H1.1-GFP depletion in the SMC was less efficient when *amiR[AIH]* was induced compared to mock-treated tissues, with 37% and 6% ovule primordia showing H1.1 persistence (*n* = 30 and *n* = 47), respectively (Figs. 3C and EV3D). To test the involvement of AIH independently, we used Cl-amidine, a known inhibitor of the mammalian citrullinase PADI4 that binds the catalytic domain (Luo et al, 2006), which is highly conserved in AIH (Fig. 3A). Cl-amidine treatment also reduced H1.1 depletion in the SMC of a large fraction of ovule primordia (39%, *n* = 56) compared to the mock treatment (Figs. 3D and EV3E). In conclusion, genetic and toxicological evidence support a model in which AIH contributes to H1.1 depletion in the SMC.

## H1 persistence in the SMC affects chromatin reorganization without impairing SMC maturation or meiosis

Next, we asked whether compromising H1.1 depletion in the SMC may affect chromatin reorganization (She et al, 2013). We focused on key markers of constitutive and facultative heterochromatin, which decrease during the SMC chromatin reorganization (She et al, 2013; Ingouff et al, 2017): we measured the relative heterochromatin fraction (RHF) and the levels of H3K27me3 and methylated CHH readers, LHP1-GFP (Exner et al, 2009) and DynaMET mCHH-Venus (Ingouff et al, 2017), respectively. SMC with a persistent H1.1$^{R57K}$-RFP signal showed higher levels of these three heterochromatin markers compared to SMCs without detectable levels of H1.1-RFP or H1.1$^{R57A}$-RFP (Fig. 4A–C). By contrast, SMC expressing the persistent H1.1$^{6xGC}$-RFP variant did not show significant changes for these markers compared to the control (Figs. 4C and EV4AB). These findings suggest a different impact of the R57K and K89R substitutions on H1.1 properties relative to chromatin reorganization in the SMC.

We then addressed whether persistence of H1.1 affects SMC differentiation and meiosis. We first analyzed SMC growth and the expression of the fate marker *KNU::nlsYFP*, two hallmarks of SMC maturation (Hernandez-Lagana et al, 2021; Tucker et al, 2003). We found that preventing H1.1 depletion by expressing either the H1.1$^{R57K}$ or the H1.1$^{6xGC}$ mutant variant compromised SMC growth, as revealed by a 3D analysis of the primordium (Figs. 4D and EV4C). In contrast, the size of companion cells increased (Fig. EV4D). Yet, *KNU::nlsYFP* expression was not altered in ovule primordia

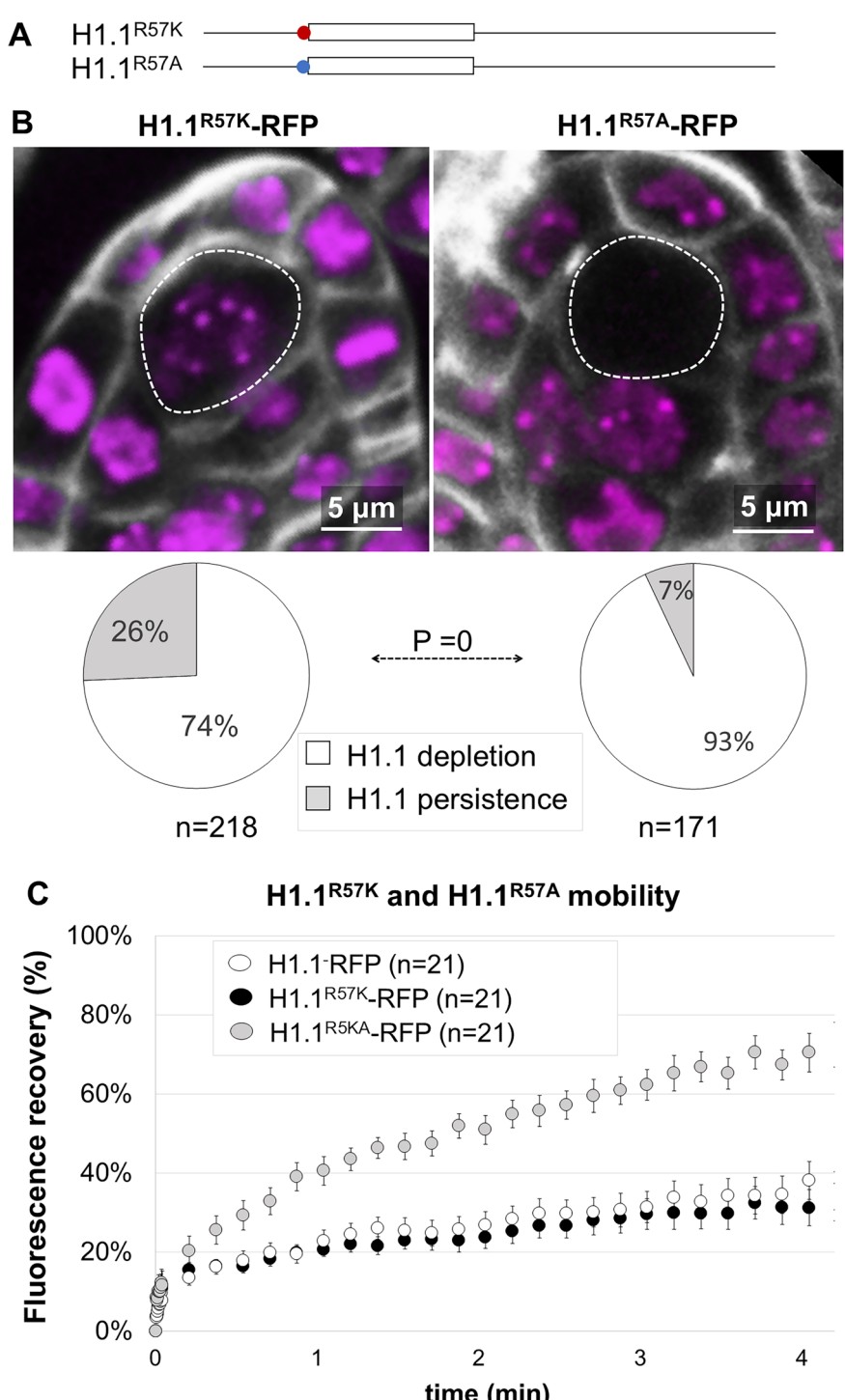

**Figure 2. An R57K substitution prevents H1.1 depletion in SMC without affecting H1.1 mobility.**

(A) Schematic representation of the H1.1 mutant variants carrying either an R57K or an R57A substitution upstream the globular domain (white box). (B) Representative images of H1.1$^{R57K}$-RFP and H1.1$^{R57A}$-RFP in ovule primordia stage 2-I at 5 dpi, showing persistence and depletion in the SMC (dotted lines), respectively (partial z-projections; RFP: magenta; Renaissance: gray). Pie charts below show the proportion of ovule primordia showing H1.1(mutant)-RFP depletion or persistence in the SMC. *n*, number of primordia scored. Dotted borders outline the SMC. *P* value, Fischer-exact test (α < 0.05). (C) Fluorescence recovery after photobleaching (FRAP) of H1.1⁻RFP, H1.1$^{R57K}$-RFP and H1.1$^{R57A}$-RFP in root nuclei from 5 days old, Dex-induced seedlings. *n*: number of analyzed nuclei; error bar: standard error to mean. See also Source data Fig. 2, Table EV1 and replicate experiments in Fig. EV2 and Source data EV2. Source data are available online for this figure.

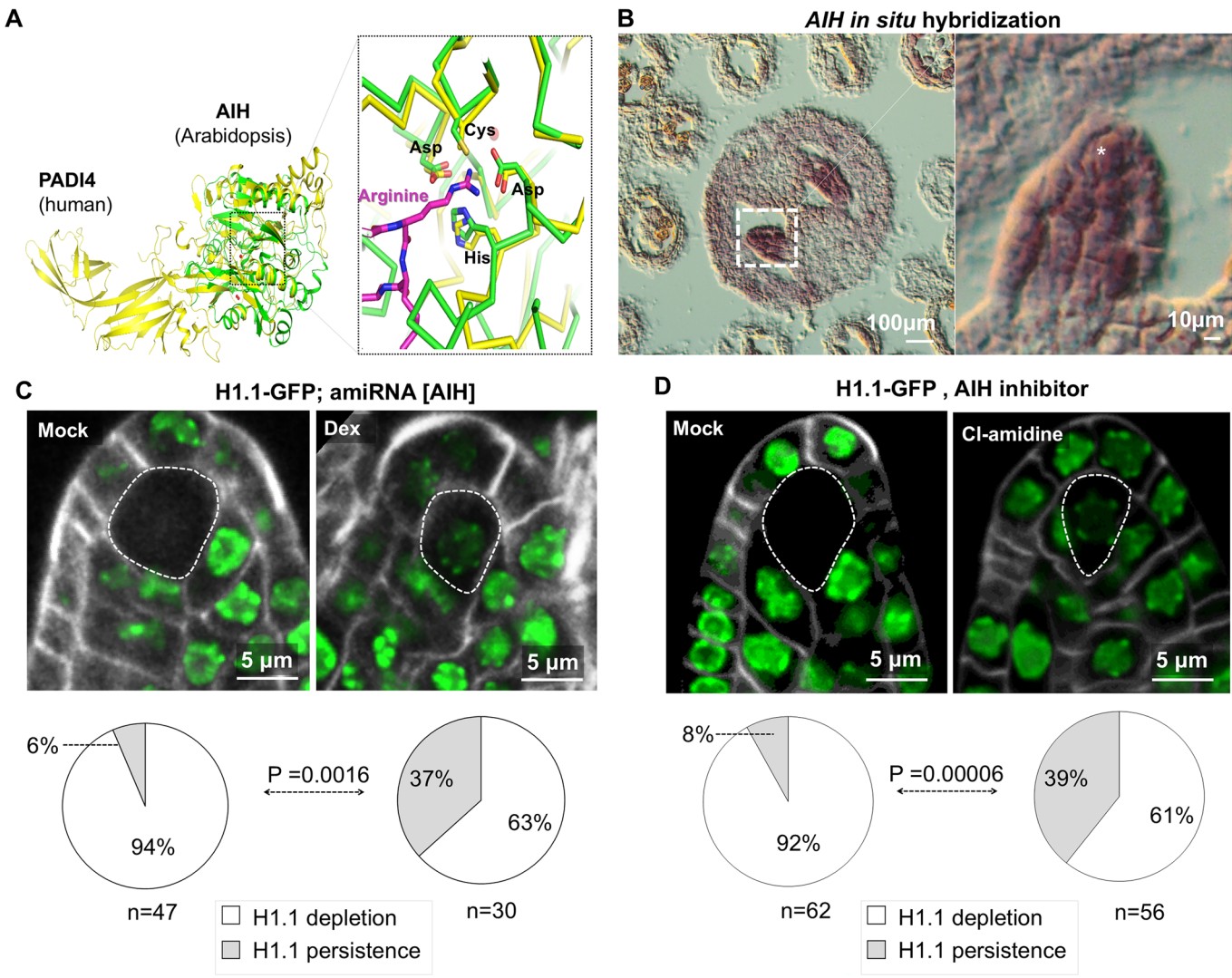

**Figure 3. Citrullinase AIH Downregulation Impairs H1.1 Depletion in SMC.**

(A) 3D protein structure of the human citrullinase PADI4 (yellow, PDB model 2DEW) and the *Arabidopsis* citrullinase AIH (green, PDB model 1VKP) showing structural homology of AIH with the catalytic head of PADI4 and a perfect conservation of the catalytic site (Asp, Cys, Asp and His, inlet). (B) Histochemical detection of *AIH* transcripts following RNA in situ hybridization on a transversal section of a flower bud stage 10 (left) revealing a specific signal in ovule primordia as shown in the box with dashed line contour and in the inset; *, SMC. (C, D) H1.1-GFP depletion in the SMC is prevented when inducing the expression of a downregulating artificial miRNA against *AIH* (*amiRNA[AIH]*, Dex versus mock (C) or when treating with the PADI4-specific enzyme inhibitor Cl-amidine (D). Images show partial projections of confocal image series showing the GFP signal (green) and Renaissance cell boundary staining (gray). Dotted borders outline the SMC. Pie charts below the images show the proportion of ovule primordia 5 dpi harboring no or persistent H1.1-GFP signal in the SMC. n, number of primordia scored. *P* value, Fischer-exact test (α < 0.05). See also Source Data Fig. 3, Table EV1 and replicate experiments in Fig. EV3 and Source data Fig. EV3. Source data are available online for this figure.

expressing either of the H1.1 mutant variants (Fig. 4E). Next, we assessed meiosis by scoring primordia expressing the ASY1-YFP marker, which is loaded onto the axis-associated chromatin in prophase I (Armstrong et al, 2002; Valuchova et al, 2020) and the occurrence of tetrads at 7dpi. Ovule primordia induced for the expression of either H1.1 mutant variants showed a normal occurrence of these meiotic markers, like the control (Fig. 4F,G). The large companion cells observed in premeiotic ovule primordia persisted at the tetrad stage (Fig. EV4E).

These results indicate that impairing H1.1 depletion in SMCs moderately affects heterochromatin reorganization and SMC growth but does not interfere with progression through meiosis.

## Premeiotic expression of the persistent variants H1.1$^{R57K}$ or H1.1$^{6xGC}$ differentially impacts gametogenesis

Female meiosis results in the production of four haploid spores, three of which degenerate. The surviving spore enlarges to become a functional megaspore. The functional megaspore will develop into the female gametophyte, or embryo sac, typically through three rounds of mitosis that are followed by cellularization to form the two female gametes and accessory cells (Skinner and Sundaresan, 2018). Given that expression of H1.1 mutant variants in SMCs does not impact meiosis, we asked whether it affects functional megaspore differentiation and embryo sac development. To this

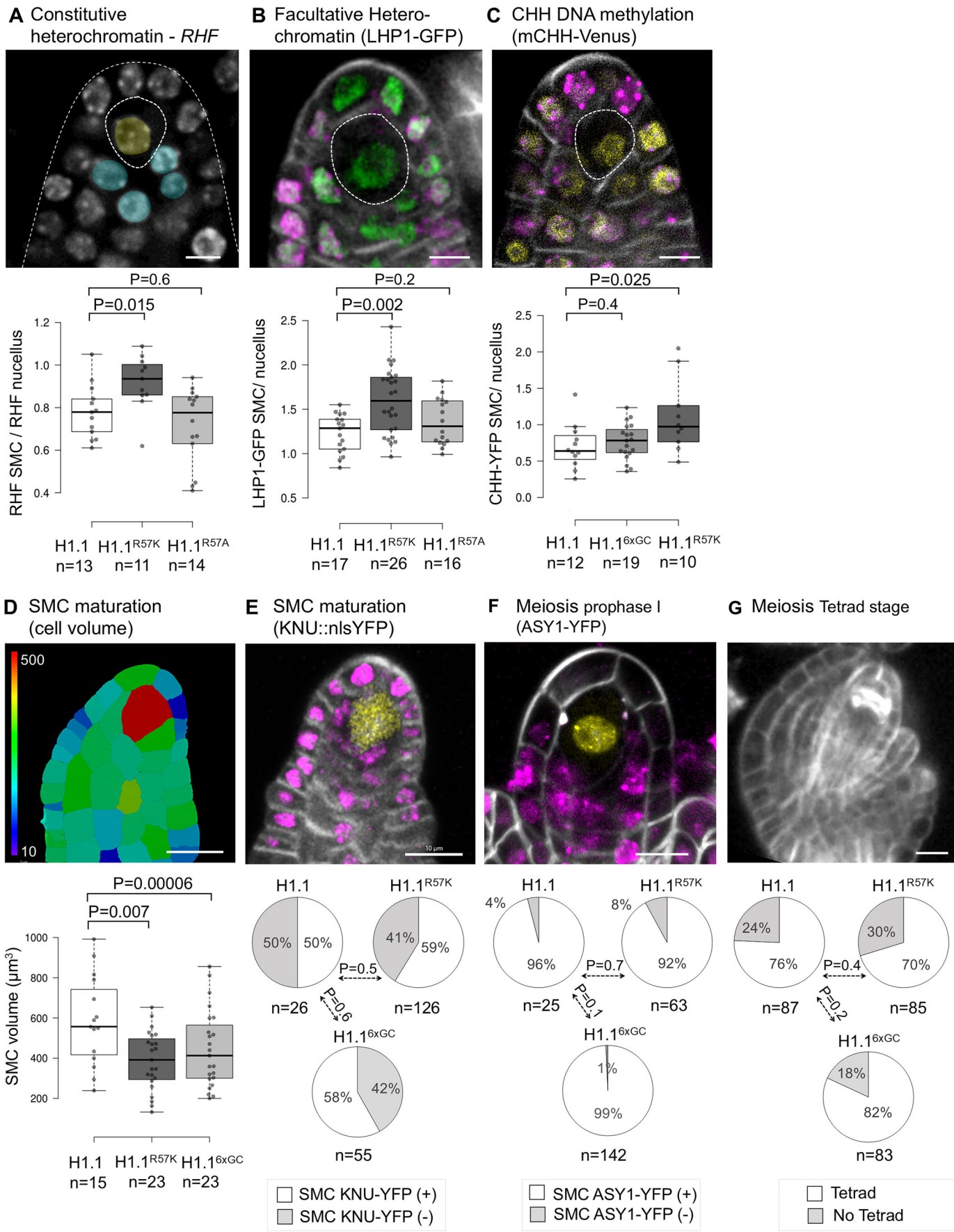

**A** Constitutive heterochromatin - *RHF*

**B** Facultative Hetero-chromatin (LHP1-GFP)

**C** CHH DNA methylation (mCHH-Venus)

**D** SMC maturation (cell volume)

**E** SMC maturation (KNU::nlsYFP)

**F** Meiosis prophase I (ASY1-YFP)

**G** Meiosis Tetrad stage

◀ **Figure 4. Inhibition of H1.1 depletion in the SMC disrupts chromatin reorganization without affecting meiosis.**

(A–C) Quantification of heterochromatin markers in the SMC of ovule primordia stage 2-I/2-II at 5 dpi in lines expressing H1.1-RFP, H1.1$^{R57K}$-RFP, H1.1$^{R57A}$-RFP or H1.1$^{6xGC}$-RFP (abbreviated H1.1, H1.1$^{R57K}$, H1.1$^{R57A}$, H1.1$^{6xGC}$, respectively). Top: representative images (confocal projections) in H1.1-RFP control lines following whole-mount propidium iodide (PI) staining (A, yellow: SMC; blue: nucellus nuclei used as references); LHP1-GFP (B, green), mCHH-Venus (C, yellow), H1.1-RFP (B, C, magenta) and Renaissance counterstaining (B, C, gray); Bottom: Graphs showing the relative heterochromatin fraction (RHF) in chromocenters (A), LHP1-GFP (B) and mCHH-Venus (C) intensities in the SMC. The measurements are expressed relative to the nucellus. (D–G) Quantification of SMC maturation in ovule primordia stage 1-II/2-I at 5 dpi (D, E) and meiosis markers in ovule primordia stage 2-III/3-I at 7 dpi (F, G) in lines expressing H1.1-RFP, H1.1$^{R57K}$-RFP, H1.1$^{R57A}$-RFP, indicated above the pie charts as in (A–C). Note that due to the progressive and variable expression of the *KNU::nlsYFP* marker, not all primordia showed a detectable signal also in the wild-type control. Top: representative confocal images in the H1.1-RFP control line following: Renaissance staining, 3D imaging and cell segmentation with cells colored according to their volume (D), live imaging of *KNU::nlsYFP* (E, yellow), ASY1-YFP (F, yellow), H1.1-RFP (E, F, magenta), Renaissance counterstaining (E–G, gray). Scale bars: 5 μm (A–C), 10 μm (D–G). Dotted borders outlines the SMC (A–C) and the ovule primordia (A). Boxplots: Center lines indicate medians; boxes span the interquartile range (25th–75th percentiles); whiskers extend 1.5× the interquartile range beyond the box limits, as computed in R. *P* values: Mann–Whitney U test (A–D), Fisher exact test (E–G). n, number of ovule primordia. See also Source data Fig. 4, Table EV1 and replicate experiments in Fig. EV4 and Source data Fig. EV4. Source data are available online for this figure.

aim we analyzed the functional megaspore-specific marker, *AKV::H2B-YFP*, a fluorescent H2B reporter driven by the *ANTI-KEVORKIAN* (*AKV*) promoter (Rotman et al, 2005; Schmidt et al, 2011). Marker expression showed similar frequencies in functional megaspores derived from ovule primordia induced for the expression of either a mutant variant—H1.1$^{R57K}$ or H1.1$^{6xGC}$—or the control H1.1 variant (Figs. 5A and EV5A). We then analyzed the embryo sacs formed by these functional megaspores, using the number of nuclei as a stage criterion. Whereas the control showed a majority of mature (FG6) embryo sacs in flower buds just before anthesis, a significant proportion of embryo sacs with earlier stages (FG2-FG4) were observed when either H1.1 mutant variant was induced (Figs. 5B and EV5B). To determine whether this was due to developmental arrest or a delay, we emasculated flowers at 9 dpi and allowed them to develop further, re-analyzing them at 11 dpi. Interestingly, in lines expressing the H1.1$^{R57K}$ variant, the proportion of mature embryo sacs at 11 dpi was no longer different than in the control line (Fig. 5B) suggesting that the abnormal embryo sacs detected at 9 dpi were delayed in their development. The same observation was made in lines expressing *amiR[AIH]* with a large fraction of delayed embryo sacs at 9 dpi that reached maturity at 11 dpi (Fig. EV5C). In contrast, a large proportion of abnormal embryo sacs persisted in lines expressing the H1.1$^{6xGC}$ variant (Figs. 5B and EV5B). The different outcomes we observed in the above experiments suggest distinct functional impacts of the R57K and K89R substitutions in H1.1 on embryo sac development.

## Premeiotic expression of either H1.1$^{R57K}$ or H1.1$^{6GC}$ variant induces seed sterility

We then asked whether the delayed embryo sacs formed following the induction of the H1.1$^{R57K}$ variant in pre-meiotic flower buds were functional. To this aim, we assessed fertility by scoring the number of plump, viable versus aborted seeds in mature siliques at 20 dpi. The control line induced for the expression of H1.1-RFP produced 87% viable seeds (*n* = 35 siliques, Fig. 5C). The line expressing H1.1$^{R57K}$-RFP, by contrast, exhibited a highly variable rate of seed abortion ranging from 0% to 95% with an average of 29% viable seeds (*n* = 40 siliques, Fig. 5C). To explain this high level of seed abortion despite the production of delayed, but morphologically normal embryo sacs, we measured pollen viability. We observed 12% aborted pollen (*n* = 323) in anthers expressing H1.1$^{R57K}$-RFP at 9 dpi compared to 3% in the control line

(*n* = 425, Fig. EV5D). This pollen abortion rate cannot explain the high frequency of aborted seeds. Hence, whether the induction of the H1.1$^{R57K}$ variant affects gametic functionality or that of sporophytic tissues remains to be investigated. In contrast, plants expressing H1.1$^{6xGC}$-RFP in premeiotic flower buds, which produce close to 90% abnormal embryo sacs had only 4.5% viable seeds on average (*n* = 40 siliques, Fig. 5C) and 67% aborted pollen (*n* = 425, Fig. EV5D). Thus, in the case of the H1.1$^{6xGC}$ variant sterility is largely due to female and male gametophytic defects although a contribution of sporophytic effects cannot be excluded. Furthermore, the induction of *amiR[AIH]* and *amiR[Cul4]* resulted in high seed sterility compared to mock controls (Fig. EV5E). This is likely because the AIH and CUL4 enzymes regulate multiple targets relevant to reproduction and not just H1. This is also consistent with the embryonic lethality of the *aih* loss-of-function mutant (*emb1873*, Meinke, 2020) and ovule and seed development defects in the *cul4* loss-of-function mutant (Dumbliauskas et al, 2011).

## Discussion

The depletion of canonical H1 linker histones is a hallmark of the somatic-to-reproductive fate transition in both plants and animals (She et al, 2013; Hajkova et al, 2008). In mice, H1.2 depletion in PGCs is critical to establish pluripotency (Christophorou et al, 2014). In the flowering plant *Arabidopsis*, the loss of H1.1 and H1.2 in male and female SMCs precedes extensive chromatin reorganization involving both structural alterations and epigenetic modifications (She et al, 2013; She and Baroux, 2015) but neither the mechanisms nor the functional implication of H1's removal were known.

### A citrullination–ubiquitination model for premeiotic H1.1 depletion in female SMCs

Previously, we proposed that the loss of H1.1 in the SMC occurs through proteasome-mediated protein degradation (She et al, 2013). Plants possess several pathways for protein degradation but ubiquitin-mediated targeting of proteins to the proteasome is prevalent (Vierstra, 1993; Moon et al, 2004). Ubiquitination involves an enzymatic cascade where ubiquitin is transferred from the E1 ubiquitin-activating enzyme to the E2 ubiquitin-conjugating enzyme, and ultimately to the target protein, either directly or

indirectly through an E3 ubiquitin ligase (Mazzucotelli et al, 2006). In this study, we demonstrate a role for the E3 ubiquitin ligase CUL4 in H1.1 degradation in the SMC, extending its previously reported functions in DNA replication, repair, and chromatin remodeling (Biedermann and Hellmann, 2011). Whether CUL4 acts directly or indirectly on H1.1, for instance via the formation of a CUL4 RING ligase (CRL4) complex, remains to be determined. CRL4s indeed target nuclear proteins and influence chromatin and chromosome function (Fonseca and Rubio, 2019). For instance, in rice, a CRL4 complex ubiquitinates the Argonaute protein MEIOSIS ARRESTED AT LEPTOTENE1 (MEL1) whose degradation in SMCs is essential for meiosis and the formation of viable microspores (Lian et al, 2021).

Presently, profiling histone ubiquitination in Arabidopsis female SMC is technically not possible. But, a previous study identified six ubiquitinated lysines in H1.1 in Arabidopsis seedling tissues (Xue et al, 2022), five in the C-terminal tail and one (K89) in the globular domain. Arginine substitutions of these residues in the H1.1$^{6xGC}$ variant prevented H1.1 degradation in the SMC, with K89 being sufficient to confer this property. These substitutions did not increase H1.1 binding affinity to chromatin, further suggesting a role for PTMs in the degradation process. We propose a model where CUL4 mediates, either directly or indirectly, the ubiquitination of K89 (and possibly other residues) for targeting H1.1 to the proteasome.

Intriguingly, a mutated H1.1 variant (H1.1$^{11xG}$, Fig. 1) with 11 substituted lysines residues in the globular domain, confirmed by sequencing the transgenes, including the K89R mutation, exhibited normal depletion in the SMC. While this suggests that K89's role in degradation requires an intact globular domain, it is unlikely that the 11 K-to-R substitutions disrupt its folding, as this domain is highly resilient to amino acid changes (Martinsen et al, 2022). The apparent lower abundance of H1.1$^{11xG}$ levels compared to other variants may suggest a lower stability of this mutant variant, masking the specific effect of K89R. At present we cannot fully explain this observation but it suggests that H1.1 degradation may be subject to versatile and redundant mechanisms involving concurrent or cooperative PTMs. In support of this hypothesis, H1.1 in leaves was found to carry a wide variety of PTMs, including crotonylation at six lysines, four of which are in the globular domain (Kotliński et al, 2016).

Furthermore, parallel efforts to identify key amino acids involved in H1.1 degradation pinpointed R57, whose charge matters. Indeed, substituting R57 with a lysine (R57K), which preserves the charge at this position, creates a variant with a similar residency time on chromatin as the wild-type variant. In contrast, neutralizing the charge by an alanine substitution (R57A) creates an H1.1 variant with a higher mobility, i.e., a shorter residency time on the chromatin as the wild-type H1.1. While the R57K substitution protects H1.1 from degradation in a significant fraction of SMC, the R57A substitution does not. In contrast, the latter seems to induce a faster depletion as suggested by the reduced fraction of SMC with residual signal at the time of scoring (Fig. EV2A). This finding indicates that H1.1 binding properties play a role in its degradation. We thus propose a model (Fig. 6) in which charge neutralization at R57 increases the dissociation rate of H1.1 from chromatin, making it more susceptible to ubiquitination-mediated degradation. This neutralization can occur, for example, through citrullination, leading to rapid H1.1

depletion in a normal context. Similarly, the R to A substitution in the H1.1$^{R57A}$ mutant results in a neutral charge leading to increased mobility, facilitating in turn degradation. In contrast, retaining a positive charge at R57, as in cells where H1.1 remains unmodified or following the R to K substitution in the H1.1$^{R57K}$ variant, lead to slower dissociation, resulting in a higher proportion of SMCs with detectable H1.1 levels at a given time. However, this substitution does not stabilize H1.1 on chromatin. The H1.1$^{R57K}$ variant exhibits similar mobility to the wild-type protein but dissociates more slowly than the citrullinated variant, eventually leading to degradation. This explains why we do not observe persistent H1.1 in all SMCs., by contrast to SMCs expressing H1.1$^{6GC}$ or H1.1$^{K89R}$ mutant variants that are protected from degradation. In line with this model involving first an increased dissociation rate by charge neutralization then degradation, the highly mobile, readily degraded H1.1$^{R57A}$ variant becomes protected from degradation when combined with a K89R substitution (Fig. EV6). Our model invokes a mechanism that neutralizes the charge at R57. Citrullination, i.e., the deimination of arginine producing a citrulline, results in the loss of a positive charge. In mouse PGCs, this modification is mediated by the citrullinase PADI4, reduces the binding affinity of H1.2 to the nucleosome and was proposed to promote H1.2 depletion in PGCs. Our findings that the plant citrullinase AIH plays a role in H1.1 depletion in the SMC support the existence of a similar mechanism in plants (Fig. 6). Notably, the arginine residue implicated in this process is located slightly upstream of the globular domain in Arabidopsis H1.1, whereas it is positioned at the start of the globular domain in the mouse H1.2 variant. This suggests that these two variants may engage different contact regions with the nucleosome.

Although the direct role of AIH in regulating H1.1 stability via arginine citrullination is the most straightforward explanation based on our findings, we cannot rule out the possibility that AIH contributes to H1.1 depletion indirectly through its role in polyamine metabolism. Polyamines have emerged as important regulators of both biotic and abiotic stress responses in plants, as well as various developmental processes, including flowering, seed development, and seedling establishment (Blázquez, 2024). Notably, the *aih* loss-of-function mutant is embryo-lethal, underscoring its essential role in development (Blázquez, 2024). Also, supporting the idea that polyamines can influence chromatin structure, spermidine deficiency—induced in the water fern *Marsilea vestita*—has been shown to impair chromatin compaction and nuclear elongation in male gametophytes (Deeb et al, 2010). Furthermore, whether additional PTM affect H1.1 mobility and as consequence, degradation rate, remains to be determined. For instance, phosphorylation is a major destabilizing PTM of H1 at the onset of S-phase and crucial for replication in animal cells (Alexandrow and Hamlin, 2005). The *Arabidopsis* H1.1 variant contains three S/TPxK motifs, which are prone to phosphorylation in the C-terminal tail (Kotliński et al, 2016), and CDC2/CDKA;1 can physically interact with H1.1 (Pusch, 2012).

The role of citrullination in chromatin regulation has been well studied in animal systems, in disease and development. Particularly, histone hypercitrullination is responsible for chromatin decondensation in activated neutrophils upon inflammatory immune response and for the production of chromatin-containing neutrophil extracellular traps (Maronek and Gardlik, 2022; Zhu et al, 2023). PAD enzymes are also associated with oncogenicity where

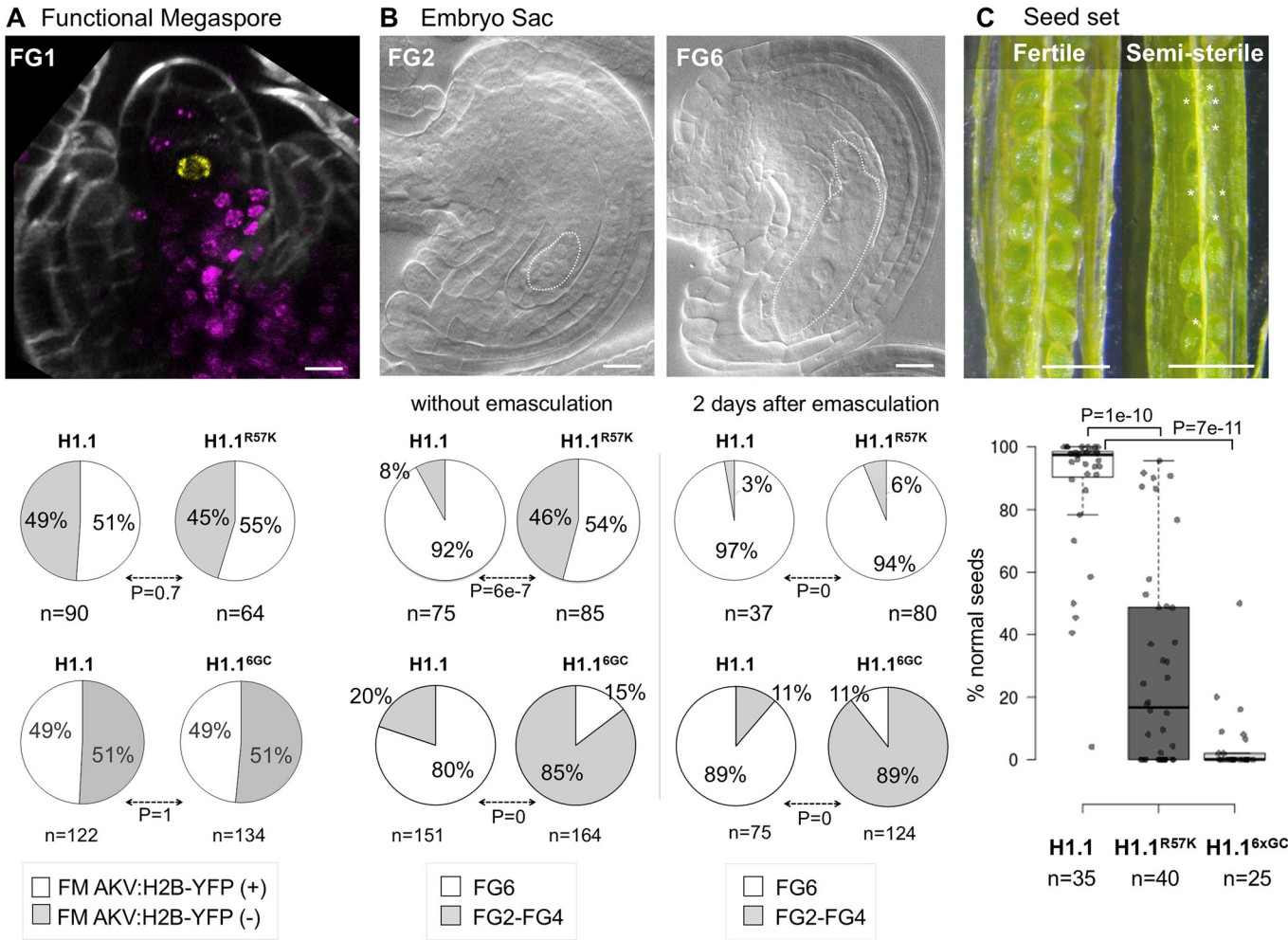

**Figure 5. H1.1 persistence in the SMC impairs gametogenesis.**

(A, B) Analysis of functional megaspore (FM) fate establishment and embryo sac development in ovules derived from primordia induced for the expression of the H1.1$^{R57K}$-RFP and H1.1$^{6xGC}$-RFP persistent variants or the H1.1-RFP control variant. Representative images of an ovule at the FG1 stage counterstained with Renaissance (gray) expressing the functional megaspore-specific marker, *AKV::H2B-YFP* (Schmidt et al, 2011) (yellow) and H1.1-RFP (magenta) at 7 dpi (A) and of cleared ovules at the FG2 and FG6 stage showing a 2-nucleate and a mature embryo sac, respectively at 9 dpi (B). Pie charts below the images show scoring of *AKV::H2B-YFP* occurrence in functional megaspores at 7 dpi (A, F1 plants) and scoring of mature embryo sacs (FG6) vs earlier stages (FG2-FG4) in ovules at 9 dpi without emasculation or at 11 dpi and after emasculation as indicated. Scoring for each mutant variant-expressing lines was compared to the control line in independent experiments. Slight differences in ovule development progression at the time of flower bud collection between experiments explain the different frequencies of AKV::H2B-YFP expression between replicates scoring in the control line. n, number of ovules scored. Dotted borders outline the embryo sac structure at different developmental stages. P value, Fischer-exact test (α < 0.05). Scale bar, 20 μm. (C) Seed set analysis in siliques 20 dpi derived from flower buds' stage 9 induced for H1.1-RFP, H1.1$^{R57K}$-RFP and H1.1$^{6xGC}$-RFP expression. Representative images of a fully fertile silique (H1.1-RFP control line), and semi-sterile silique (H1.1$^{R57K}$-RFP mutant line) at 20 dpi and scoring of normal seeds (box plots) for all three genotypes as indicated. *, infertile ovules. n, number of siliques scored (ca 50 seeds p. silique). Scale bar: 1 mm. P values, Mann–Whitney U test. Boxplots: Center lines indicate medians; boxes span the interquartile range (25th–75th percentiles); whiskers extend 1.5× the interquartile range beyond the box limits, as computed in R. See also Source data Fig. 5, Table EV1 and replicate experiments in Fig. EV5 and Source data Fig. EV5. Source data are available online for this figure.

their ectopic activation leads to chromatin decondensation, ectopic histone citrullination and deregulation of gene expression (Zhu et al, 2021; Zheng et al, 2020). Yet, histone citrullination also plays a role in development. Notably, the citrullination of H1 (Christophorou et al, 2014) but also of H3 and H4 (Xiao et al, 2017) are necessary for activating pluripotency genes in PGCs. After fertilization, selective citrullination of H3 and H4 is necessary for embryonic genome activation and proper gene expression (Zhang et al, 2016).

By contrast, much less is known about citrullination in plants. The catalytic domain of the plant AIH citrullinase is highly conserved with its mammalian PAD4 counterpart: it shares the four key catalytic amino acids and exhibits similar inhibitor sensitivity, indicating that citrullination may play a fundamental role in plants too. Indeed, protein citrullination has recently been identified as a significant post-translational modification in response to cold stress in cell cultures (Marondedze et al, 2021). Our discovery that AIH is required for H1.1 depletion in SMCs opens new avenues for

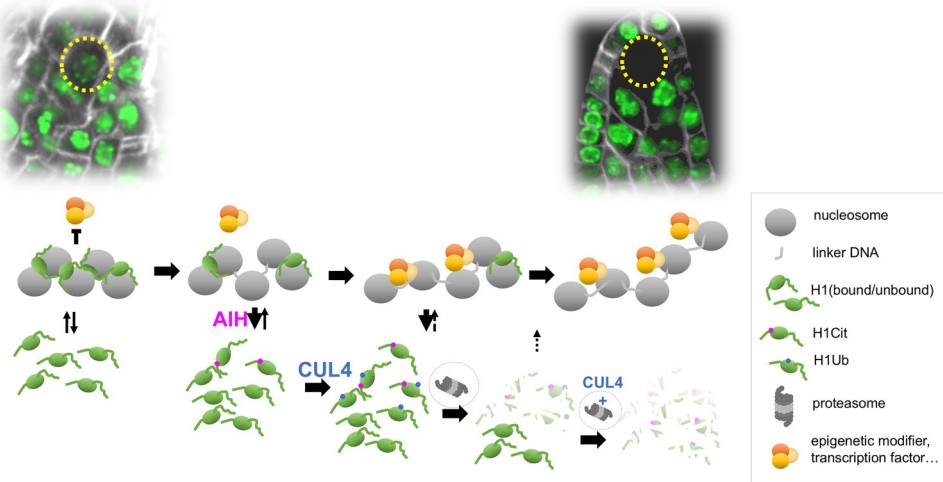

**Figure 6.  A two-step dissociation-degradation model of H1.1 depletion in the SMC.**

Based on our findings, we propose a two-step model explaining the depletion of H1.1 from the SMC chromatin. First, the association of H1.1 with chromatin is electrostatically destabilized due to charge neutralization by the citrullinase AIH, particularly at residue R57, while other target residues are not excluded. This facilitates dissociation from chromatin and increases the pool of unbound H1.1 proteins susceptible to ubiquitination by CUL4 and subsequent proteasome-mediated degradation. As H1.1 binding dynamics to the nucleosomes are altered, the progressive degradation of H1.1 depletes the available pool for binding. Meanwhile, reduced transcription and/or translation in the SMC, broad or gene-specific, prevents the replenishment of the H1.1 pool. As a result, chromatin becomes accessible to epigenetic modifiers, transcription factors, or both, to render the cell competent for the post-meiotic formation of the embryo sac.

investigating the role of histone citrullination in plant chromatin dynamics.

A key question arising from our findings is the cell specificity of H1.1 degradation. Both AIH and CUL4 are broadly expressed in ovule primordia, a pattern that does not explain the observed cell-specific degradation. One possibility is that these enzymes might be catalytically regulated by a cofactor conferring cell specificity. For example, mammalian PADI4 activity is influenced by $Ca^{2+}$ availability (Arita et al, 2004). Although calcium showed undetectable levels in male SMCs in *Nicotiana tabacum* and *Torenia fournieri* (Ge et al, 2008), no studies have yet been conducted on calcium availability in ovule primordia or in female SMCs. Also, considering the intricate interplay between calcium, reactive oxygen species, and the cellular availability of free oxygen (Görlach et al, 2015), as well as the recent discovery that hypoxia regulates protein degradation and stem cell fate in the shoot apical meristem (Weits et al, 2019) and can induce meiotic fate acquisition in maize (Kelliher and Walbot, 2012), exploring the role of hypoxia in SMC differentiation—particularly in relation to chromatin remodeling—presents an intriguing avenue for future research, which will require solving challenges in experimental approaches. A second possibility is that of an SMC-specific AIH isoform encoded by one of the distinct splice variants (Fig. EV3B). Finally, an interesting alternative hypothesis exists that contrasts with the notion of specific activity of AIH and CUL4. Instead, it posits H1.1 dissociation-degradation as the default process in all cells of the primordium corresponding to the broad domain of *AIH* and *CUL4* expression, but an SMC-specific lack of transcriptional and/or translational compensation prevents the replenishment of the H1.1 pool in these cells. With an average polyA tail of ~55–60 nt and a half-life of about 1 h in vegetative tissues, *H1.1* transcripts do not exhibit distinctive features (Jia et al, 2022; Szabo et al, 2020). However, this short lifetime of *H1.1* transcripts, in combination

with a relative transcriptional quiescence of the SMC (She et al, 2013) may be sufficient to prevent the replenishment of H1.1 on chromatin. Whether translational inhibition also occurs in the SMC, targeting specific transcripts such as those encoding H1.1, or if it is part of a broader mechanism—similar to what is seen in animal PGCs (Oulhen et al, 2017)—remains to be determined. Interestingly, premeiotic spikelet and male meiocytes in prophase I of rice are enriched in phased secondary small interfering RNAs (phasiRNAs) involving MEL1 (Komiya et al, 2014) and bearing the potential for translational inhibition (Jiang et al, 2020; Liu et al, 2020). Future studies could explore the repertoire of small and antisense RNAs in the female SMCs of *Arabidopsis* that could influence transcription and translation. This model, in which H1.1 undergoes rapid turnover throughout the primordium but is not replenished in the SMC due to stalled transcription and/or translation may also apply to other chromatin constituents such as H2A.Z and H3.1/HTR13 that are specifically lost in the SMC (She et al, 2013; Hernandez-Lagana et al, 2021). This opens the exciting question of which process(es) specifically reduce transcriptional and/or translational activity in the SMC.

## Implications of premeiotic H1.1 clearance for gametogenesis

H1 depletion in mouse PGCs and subsequent chromatin reorganization are essential for establishing pluripotency in the mammalian germline (Christophorou et al, 2014). Although similar events were known to occur during SMC formation in *Arabidopsis*, marking the transition from somatic to reproductive fate (She et al, 2013; She and Baroux, 2014), the precise role of H1 depletion remained unknown. In *Arabidopsis*, linker histones interplay with the deposition and maintenance of various epigenetic marks, including DNA methylation, histone

methylation—notably H3K27me3—, histone acetylation, and drives heterochromatin formation (Wierzbicki and Jerzmanowski, 2005; Zemach et al, 2013; Rutowicz et al, 2019; Choi et al, 2020; Bourguet et al, 2021; He et al, 2024). Notably, depletion of linker histones at the onset of SMC differentiation precedes the gradual decrease in constitutive heterochromatin (chromocenters) and in H3K27me3 (She et al, 2013) as well as a decrease in mCHH (Ingouff et al, 2017), compared to surrounding cells. SMC expressing H1.1[R57K] show moderately—but significantly—elevated levels of these chromatin features as measured by the relative heterochromatin fraction and by fluorescently tagged H3K27me3 and mCHH readers compared to wild-type. By contrast, the expression of H1.1[6xGC] led to little or no effects on chromatin reorganization. These distinct outcomes can be explained through our model (Fig. 6) as follows: the slower dissociation rate of H1.1[R57K] mutant, compared to the citrullinated wild-type H1.1 variant (H1.1cit), leads to prolonged chromatin residency. This extended binding time may partially hinder or delays hetero-chromatin decondensation, resulting in increased RHF levels as measured. Similarly, it may partially hinder the global reduction in H3K27me3 and mCHH which in turn leads to elevated levels of their respective reporters. By contrast, the H1.1[6xGC] variant retains the ability to be citrullinated at R57, resulting in a higher dissociation rate and shorter chromatin residency time—similar to H1.1cit—which may be favorable to chromatin reorganization. These findings highlight the role of H1.1 removal—initiated by its increased dissociation rate and culminating in degradation—in promoting chromatin reorganization in SMCs, a process that likely creates a window of opportunity for epigenetic reprogramming during the somatic-to-reproductive fate transition.

Given the extent of chromatin reorganization following H1.1 and H1.2 depletion during SMC differentiation (She et al, 2013; She and Baroux, 2015), the finding that impaired H1.1 degradation is not essential for meiosis was unexpected. H1.1 depletion influences SMC growth, possibly through the de-repression of growth factors that remains to be identified, but this does not appear to be critical for SMC functionality. Whether H1.2 depletion, instead, plays a decisive role in SMC differentiation and the execution of meiosis remains to be investigated. This would hint at distinct functions between these two variants that are structurally very similar and follow a similar expression pattern (Rutowicz et al, 2015; Kotliński et al, 2017). Rather than serving a pre-meiotic function, H1.1 depletion in female SMCs appears to be crucial for establishing post-meiotic developmental competence required for female gamete formation.

Functional megaspores derived from SMCs with impaired H1.1 depletion exhibited mitotic defects, resulting in delayed or arrested embryo sac development at the 2- to 4-nucleate stage. In mature flowers, none of the embryo sacs examined displayed a fully developed, cellularized structure containing gametes and accessory cells. This finding suggests a failure in the expression of one or several components controlling mitosis, nuclear migration, cell growth, or a combination thereof. In Arabidopsis, H1 variants are not deposited at specific loci but are instead broadly distributed across the genome, with a notable enrichment at lowly expressed genes (Rutowicz et al, 2015). As such, the impact of H1 removal in the SMC on genes involved in embryo sac formation and gametogenesis remain unclear. In mammals, H1 variants are similarly distributed across the genome, and their reduced stoichiometry results in gene expression changes associated with local chromatin decompaction, with levels of

H3K27me3 and H3K36me3 decreasing and increasing, respectively (Willcockson et al, 2021). In Arabidopsis, H1 depletion moderately affects gene expression and chromatin accessibility, particularly at loci targeted by the Polycomb Repressive Complex 2 (PRC2) (Rutowicz et al, 2019; Teano et al, 2023). It also broadly influences accessibility to the chromatin remodeler DDM1, which plays a key role in DNA methylation (Zemach et al, 2013). Determining whether H1.1 removal in the SMC influences gene expression globally or at specific loci by modulating chromatin accessibility to chromatin modifiers like DDM1 and PRC2 will be an intriguing, yet challenging, avenue for future investigation. Alternatively, and not exclusively, a model in which H1.1 competes with transcription factors for DNA binding could offer a mechanism to control a subset of embryo sac-specific genes. This hypothesis is supported by the observation that H1 depletion in Arabidopsis primarily impacts NAC target genes among diurnally regulated genes (Rutowicz et al, 2025). In apple, a linker histone variant was also shown to act as a direct transcriptional regulator of metabolic genes (Hu et al, 2025). Thus, exploring whether H1.1 depletion affects a specific set of loci enriched with DNA-binding motifs, through cell-specific transcriptome profiling in the embryo sac, would provide valuable insights.

Interestingly, the H1.1[R57K] mutant, affecting H1.1 dissociation rate from chromatin in the SMC, had a different impact on embryo sac development than the H1.1[6xGC] mutant, which abolishes degradation. In the former, embryo sac development was delayed, while in the latter, it was arrested. These findings may be explained by a scenario in which, although H1.1[R57K] is degraded less efficiently, a sufficient fraction is still removed, allowing activation of key target loci and permitting mitosis in the embryo sac to proceed, albeit more slowly. In contrast, the continuous presence of the H1.1[6xGC] mutant variant, which cannot be degraded, compromises transcriptional reprogramming more severely, impairing on mitosis in the embryo sac.

We propose that pre-meiotic H1 depletion contributes to reshaping the epigenetic landscape of the SMC, particularly in relation to DNA methylation and H3K27me3—two key epigenetic marks known to depend on H1 abundance. This remodeling likely facilitates the activation of the post-meiotic transcriptional program in the functional megaspore, which is essential for embryo sac development. Future studies employing transcriptome profiling, assisted by laser capture microscopy to achieve cell specificity in functional megaspores derived from wild-type SMCs or those impaired in H1.1 depletion, will help addressing this question.

The parallel with H1 depletion in mouse primordial germ cells (PGCs), which is essential for establishing pluripotency (Christophorou et al, 2014), is compelling—especially considering that the functional megaspore also gives rise to a multicellular embryo sac composed of distinct cell types. These similarities suggest a potential evolutionary convergence between plant and animal systems, where histone H1 removal and associated chromatin remodeling play a critical role in enabling cell fate transitions during reproductive development (Hajkova et al, 2008; She et al, 2013).

Finally, future investigations should explore whether H1 citrullination and ubiquitination also influences the binding dynamics, abundance and stoichiometry of all three H1.1, H1.2 and H1.3. Several phenotypes resulting from H1 depletion have been described in Arabidopsis, such as altered development (flowering time, lateral root and root hair abundance, stomatal patterning, Rutowicz et al, 2019) and environmental responses (to drought and low light, salt stress, heat stress, and defense priming, Rutowicz et al, 2015; Liu et al,

2021; Sheikh et al, 2023; Perrella et al, 2024). Yet, the specific role of H1 variant stoichiometry—i.e., the delicate balance in abundance and genomic distribution of all three variants in a cell and tissue-type-specific manner during development and biotic and abiotic stress responses—has yet to be explored.

# Methods

### Reagents and tools table

| Reagent/Resource | Reference or Source | Identifier or Catalog Number |
|---|---|---|
| **Experimental models** | | |
| Arabidopsis thanliana *Colombia 0* | NASC | N70000 |
| pKNU::nlsYFP Col0 background | Tucker et al, 2003 | |
| pASY1::ASY-YFP, Col0 background | Valuchova et al, 2020 | |
| pAKV::H2B-YFP, Col0 background | Rotman et al, 2005 | |
| pH1.1::H1.1-GFP, pH1.2::H1.2-ECFP, *3h1* background | She et al, 2013 | |
| **Recombinant DNA** | | |
| pDonor pRPS5a::LhGR2_GUS::pOP6::attL1-cddb-attL2 vector | Samalova et al, 2019 | |
| pH1.1::H1.1-GFP | Rutowicz et al, 2015 | |
| pH1.2::H1.2-GFP | Rutowicz et al, 2015 | |
| pEntry, attL1-H1.1-R57K-RFP-attl2 in pMA-RQ (ampR) | This study | |
| pEntry, attL1-H1.1-R57A-RFP-attl2 in pMA-RQ (ampR) | This study | |
| pEntry, attL1-H1.1-R79K-RFP-attl2 in pMA-RQ (ampR) | This study | |
| pEntry, attL1-H1.1-R79A-RFP-attl2 in pMA-RQ (ampR) | This study | |
| pEntry, attL1-H1.1-K89R-RFP-attl2 in pMA-RQ (ampR) | This study | |
| pEntry, attL1-H1.1-6GC-RFP-attl2 in pMA-RQ (ampR) | This study | |
| pRPS5a::LHGR2::popOn2 (pRPS5a>>) | Craft et al, 2005; Samalova et al, 2019 | |
| pRPS5a»amiRNA[CUL4] | This study | |
| pRPS5a»H1.1-6xGC-RFP | This study | |
| pRPS5a»H1.1-RFP | This study | |
| pRPS5a»_H1.1-R57K-RFP | This study | |
| pRPS5a»H1.1-R57A-RFP | This study | |
| pRPS5a»amiRNA[AIH] | This study | |
| **Antibodies** | | |
| primary anti-CUL4 IgG antibody | https://www.phytoab.com/cul4%20antibody | PhytoAB Cat#PHY0850A |
| secondary HRP-conjugated anti-rabbit IgG antibody | Amersham ECL Rabbit IgG, HRP-linked whole Ab, https://www.cytivalifesciences.com | Amersham™ NA934 |
| **Oligonucleotides and other sequence-based reagents** | | |
| *AIH* Probe synthesis forward primer | 5′-CGCTTTGGAACGAA TTCC-3′ | |
| *AIH* Probe synthesis reverse primer | 5′-TAATACGACTCAC TATAGGGCCGAAAG AGCTTCCACAG-3′ | |
| **Chemicals, Enzymes and other reagents** | | |
| BACTO™ Agar | BD BACTO™ | SKU:214030 |
| Triton™ X-100 | Sigma-Aldrich Merck KGaA, Germany | 9036-19-5 |
| Murashige and Skoog salts | Carolina | 19-5700 |

| Reagent/Resource | Reference or Source | Identifier or Catalog Number |
|---|---|---|
| Solbac | Andermatt Biocontrol Suisse AG | Item No. 933C |
| Gateway™ LR Clonase™ II Enzyme mix | Thermo Fisher Scientific | Catalog number 11791020 |
| Dexamethasone | Sigma-Aldrich Merck KGaA, Germany | 265005 |
| Dimethyl sulfoxide | Sigma-Aldrich Merck KGaA, Germany | CAS Number:67-68-5 |
| Epoxomicin | LUCERNA-CHEM | Cat Nr HY-13821 |
| Tween™ 20 Surfact-Amps™ Detergent Solution | Thermo Fisher Scientific | Cat Nr 85113 |
| Cl-amidine | Sigma-Aldrich Merck KGaA, Germany | Calbiochem® 506282 |
| Pierce™ BCA Protein Assay Kits | Thermo Fisher Scientific | Cat Nr 23225 |
| SDS page gel | 4–20% Mini-PROTEAN® TGX™ Precast Protein Gels, Bio-Rad Laboratories | #4561094 |
| Ponceau red | Ponceau S solution, PanReac AppliChem | Cat#A2935,0500 |
| Clarity Max Western ECL Substrate | Bio-Rad Laboratories | Bio-Rad 1705062 |
| VECTASHIELD Antifade Mounting Medium | Vector Laboratory, USA | H-1000-10 |
| Invitrogen™Propidium Iodide | Thermo Fisher Scientific | Cat Nr P1304MP |
| DIGOXIGENIN (DIG) RNA Labeling Kit | DIG RNA Labeling Mix, Sigma-Aldrich Merck KGaA, Germany | Cat.No. 11277073910, |
| T7 RNA polymerase | Sigma-Aldrich Merck KGaA, Germany | 10881767001 |
| Histoclear | HistoChoice® Clearing Agent, Sigma-Aldrich Merck KGaA, Germany | H2779 |
| alkaline phosphatase (AP), | Anti-Digoxigenin-AP, Fab fragments, Sigma-Aldrich Merck KGaA, Germany | 11093274910 |
| BCIP/NBT Color Development Substrate, | 5-bromo-4-chloro-3-indolyl-phosphate/nitro blue tetrazolium, Promega, Switzerland | Catalog Number: S3771 |
| **Software** | | |
| Fiji | https://fiji.sc/ | |
| Imaris software (Bitplane, Switzerland | https://imaris.oxinst.com/ | |
| PyMOL | Chaudhari and Li, 2015 | |
| **Other** | | |
| Trans-Blot® Turbo™ Transfer System | Bio-Rad Laboratories | 1704150EDU |
| ChemiDoc Imaging System (Bio-Rad) | Bio-Rad Laboratories | 12003153EDU |
| Confocal, Leica SP5-R | Leica Microsystems, Germany | |
| Confocal, Stellaris | Leica Microsystems, Germany | |
| Widefield Microscope, Leica DMR | Leica Microsystems, Germany | |
| Flexacam C3 12 MP Microscope Camera | Leica Microsystems, Germany | |
| Confocal Leica SP8-R, inverted | Leica Microsystems, Germany | |
| Tissue processor for RNA ISH | Stirrer bar Leica ASP200/ 300 S, Biosystems Switzerland AG | 14047643630 |

## Plant growth conditions

*Arabidopsis* seeds were sterilized in freshly made 0.03% bleach and 0.05% Triton X-100 for 10 min, washed three times in sterile water,

briefly incubated in 70% EtOH and washed once before sowing on the germination medium consisting of 0.5xMurashige and Skoog salts (MS, Carolina 19-5700, USA), 10% (w/v) Bactoagar, pH 5.6. Seeds were stratified 2 to 4 days at 4 °C before being transferred to a growth incubator (Percival) with long day conditions (16 h light [120 µE m$^{-2}$s$^{-1}$] at 21 °C and 8 h dark at 16 °C). Ten days-post-germination, seedlings at the 2–4 leaves stage were transplanted to soil, covered with a layer of sand and imbibed with Solbac (Andermatt Biocontrol, Switzerland) for pest biocontrol. Plants were cultivated in a growth chamber with controlled conditions under a long-day photoperiod (16 h light) at 18–20 °C and 38% to 59% rF of humidity.

## Generating construct and transgenic lines

The artificial micro-RNAs (amiRNA) targeting CUL4 (At5g46210) or AIH (At5g08170) were designed with the wmd3 database (Meyers and Green, 2010) and comprised the following sequences: 5'-ATGC*TCATGATTCGTAGTATA-3' and 5'-ACGC*-GAGCCGTTCATTAATTA*-3', respectively, with mismatches to the original sequence indicated by an *. The amiRNAs and the H1.1 mutant variants modified for specific amino acid substitutions were synthesized by Genscript (GenScript Biotech Corporation, genscript.com) and introduced into the *BAR* gene-containing donor vector pRPS5a::LhGR2-GUS::pOP6::attL1-cddb-attL2 vector (Samalova et al, 2019) via Gateway cloning (Thermo Fischer Scientific, USA) according to the manufacturer's recommendation. The resulting constructs, abbreviated as *amiRNA[Cul4]* and *amiRNA[AIH]*, were transformed via *Agrobacterium tumefaciens* (GV3101) into a *pH1.1::H1.1-GFP; pH1.2::H1.2-ECFP; 3h1 Arabidopsis* line. This line was created by introgression of *pH1.1::H1.1-GFP* and *pH1.2::H1.2-ECFP* into the triple H1 mutant described, *3h1* (She et al, 2013) by crossing and selection. The *pH1.1::H1.1-GFP* line was described (Rutowicz et al, 2015) and the *pH1.2::H1.2-ECFP* line was created as described for the *pH1.2::H1.2-GFP* line (Rutowicz et al, 2015) albeit replacing the *GFP* sequence with the *ECFP* sequence (Clontech).

The constructs encoding RFP-tagged H1.1 variants containing codon changes for amino acid substitutions were synthesized by Genscript (GenScript Biotech Corporation, genscript.com), introduced into the same donor vector via Gateway cloning as described above and transformed into the *Arabidopsis* Col-0 accession. Positive T1 plants were identified based on selection for the BAR gene in vitro (0.5x MS medium supplemented with 50 µM phosphinothricin) followed by a GUS reporter assay on the 3rd or 4th true leaf after treatment with 10 µM dexamethasone (see below).

## Histochemical detection of GUS reporter activity (GUS reporter assay)

To select transgenic lines with a functional construct we verified each resistant T1 plants for the expression of the *uidA* reporter gene encoding a β-glucuronidase (GUS) and present in the responder cassette (Craft et al, 2005; Samalova et al, 2019). For this, one cauline leaf per seedling was incubated with the reaction buffer (0.1% Triton X-100, 10 mM EDTA, 2 mM ferrocyanide, 2 mM ferricyanide, 100 mM Na$_2$HPO$_4$, 100 mM NaH$_2$PO$_4$, 2 mg/mL X-Gluc) in 48-multiwell plates, vacuum infiltrated for 5 min and stained for 6–8 h at 37 °C.

## Dexamethasone treatments

*Arabidopsis* lines containing the dexamethasone (Dex)-inducible constructs were induced as described (Schubert et al, 2022). Briefly, flower buds were delicately opened with dissecting needles under the stereomicroscope to allow for a good exposure of the carpel, then gently brushed, using a fine paint brush with soft hairs, either with the induction solution (10 µM Dexamethasone [Sigma-Aldrich Merck KGaA, Germany], 0.1% DMSO [Sigma-Aldrich Merck KGaA, Germany], 0.01% Silwet-L77, ddH$_2$O) or with a mock solution (0.1% DMSO, 0.01% Silwet-L77, ddH$_2$O). Flower buds or siliques were collected several days post-induction (dpi) depending on the experiment. The preparation of this material for microscopical analyses is described further below.

## Epoxomycin and Cl-amidine treatments

For epoxomicin treatment, intact inflorescences were collected and flower buds older than stage 10 (Smyth et al, 1990; Yu et al, 2020) were removed. The remaining inflorescences were immersed in an epoxomicin-containing solution (5 µM epoxomicin (LUCERNA-CHEMÒ), 0.01%Silwet-77L, 0.01%Tween-20, ddH$_2$O) in a 48-multiwell plate and incubated in a growth incubator (Percival, long days, 21 °C) for 2 days before sample preparation for imaging. For Cl-amidine treatment, young flower buds were treated with 1 µM Cl-amidine (CalbiochemÒ) supplemented with 0.01% Silwet-L77 or with a mock solution (0.01%DMSO, 0.01% Silwet-L77) the same way as for the Dex treatments described above.

## Western blot and quantification

For protein extraction, ~100 mg of 10-day-old *Arabidopsis* seedlings were harvested and ground using a Retsch homogenizer in liquid nitrogen. The ground tissue was resuspended in 400 µL of SDS lysis buffer (40 mM Tris-HCl pH 7.5, 10% glycerol, 5 mM MgCl$_2$, 4% SDS). The samples were incubated at room temperature for 10 min under gentle shaking at 400 rpm (Eppendorf Thermo-Mixer) to solubilize the tissue evenly in the SDS lysis buffer. Following a centrifugation at $13,000 \times g$ for 5 min the supernatant was collected, and centrifugated again at $13,000 \times g$ for 5 min. The final supernatant was collected for protein quantification using the Pierce BCA Protein Assay kit (Thermo Fisher Scientific), following the manufacturer's protocol with 10 µL of the sample/standard. All sample concentrations were normalized by diluting with SDS lysis buffer. The freshly prepared protein samples were then mixed with 1 µL 5x loading buffer (250 mM Tris-HCl pH 6.8, 5% SDS, 50% glycerol, 1 mg bromophenol blue, 5% β-mercaptoethanol) and boiled for 10 min at 95 °C.

For immunoblotting, 10 µg of protein extract was loaded for each sample onto an SDS page gel (Mini-PROTEAN TGX™ Precast Gels, Bio-Rad, #4561094). Electrophoresis was performed at 90 V, followed by the transfer of proteins onto a nitrocellulose membrane using the Trans-Blot Turbo Transfer System (Bio-Rad) at 2.5 A constant, 25 V for 20 min. The membrane was stained 1 min with Ponceau red (PanReac AppliChem, Cat#A2935,0500), before acquiring an image with the ChemiDoc Imaging System (Bio-Rad). The membrane was then washed with ddH$_2$O and blocked with TBS-T (1xTBS, 1% Tween) blocking buffer containing 5% bovine serum albumin (BSA) with shaking for at least 60 min. The

membrane was sealed in a plastic pouch and incubated overnight at 4 °C on a shaker with 3 mL of primary anti-CUL4 antibody (PhytoAB Cat#PHY0850A) at 1:2000, all diluted in blocking buffer.

Following the primary antibody incubation, the membrane was washed three times for 10 min each with TBS-T before incubation with the secondary HRP-conjugated anti-rabbit IgG antibody (Amersham NA934) diluted at 1:10,000 in TBS-T supplemented with 5% milk, for 1 h on a shaker. The membrane was washed again three times for 10 min each with TBS-T then incubated for 1 min with 500 μL of Clarity Max Western ECL Substrate (Bio-Rad 1705062) and imaged using the ChemiDoc Imaging System (Bio-Rad). The relative intensity mean was calculated for each sample as the ratio between the intensity mean of the predicted band around CUL4 ~ 91KDa and the intensity mean of the major band on the Ponceau staining ~70 kDa. The ratio was then expressed relative to Line4-Rep2 set to "1". Measurements were done using Fiji on 8 bit greyscale inverted images using ROI of the same size.

### 3D Confocal imaging of whole-mount ovule primordia

Carpels freshly dissected from flower buds were delicately opened using insulin needles to remove the carpel walls and expose the ovule primordia, and mounted in Renaissance solution (final concentrations: 1:2000 Renaissance; 10% glycerol; 0.05% DMSO; 0.1%Triton X-100 in 1x PBS (modified from Musielak et al, 2015) on a Superfrost™ slide (Thermo Fisher Scientific) and covered with a $18 \times 18$ mm coverslip, #1.5 thickness (Assistant, Germany) and immediately imaged. Fixed, cleared ovules stained with propidium iodide in acrylamide pads (see below) were mounted in Vectashield supplemented with Propidium Iodide (H1300-10, Vector Laboratory, USA) and covered with a $18 \times 18$ mm coverslip, #1.5 Thickness (Assistant, Germany). Serial image acquisition was carried out using confocal microscopy (CSLM Leica SP5, or Stellaris, Leica Microsystems, Germany) with an HC PL APO 63x/1,30 GLYC CORR CS2 objective lens, using the fluorophore's optimal excitation laser, an emission window of circa 40–60 nm centered on the emission maximum, using a resonance scanning mode (8000 Hz), photon counting mode for detection, and background free HyD detectors, with signal accumulation per line to optimize the pixel intensity distribution. Images were acquired with voxel size of 0.08–0.1 μm x,y,z dimension.

### Microscopy-based scoring of the H1 depletion phenotype in female SMCs

The H1.1 depletion phenotype in SMCs was analyzed in 10–30 ovule primordia per flower bud, from 2–3 independent flower buds per plant and 2–3 replicate plants under confocal live scanning (see above). Only ovule primordia at the relevant stage and showing clear signal and structure were scored. The control and mutant plants, Mock or Dex treated plants were analysed blind. Three categories were considered: (i) Full depletion: no detectable signal in the SMC; (ii) Partial depletion: detectable signal, usually in chromocenters; (iii) Persistence: clear signal in both euchromatin and chromocenters.

### Microscopy-based scoring of ovule stage-specific markers

Ovule primordia expressing the SMC, meiosis and gametophyte-specific markers *KNU::nlsYFP* (Tucker et al, 2003), *ASY1::ASY1-*

*YFP* (Valuchova et al, 2020) and *AKV::H2B-YFP* (Schmidt et al, 2011; Rotman et al, 2005), respectively, were prepared as described for microscopy analysis. Ovules were scored under live imaging for presence/absence of signal at developmental stages and in genetic backgrounds indicated in the Figures and associated data.

For tetrad analysis in ovules, the samples were mounted in Renaissance solution (final concentrations: 1:2000 Renaissance; 10% glycerol; 0.05% DMSO; 0.1%Triton X-100 in 1x PBS (modified after Musielak et al, 2015). The presence of normal versus abnormal tetrads was scored under live, 3D imaging at stages and in genetic backgrounds indicated in the Figures and associated data.

### Analysis of embryo sac development

Embryo sac development was analyzed by identifying the developmental stages as described (Schneitz et al, 1995), either using the *AKV::H2B-YFP* reporter line (Schmidt et al, 2011) introgressed into the mutant backgrounds, or by tissue clearing as indicated in the relevant Figures. For tissue clearing, inflorescences were fixed in acetic acid: EtOH solution (3:1) and incubated for at least 4 h at room temperature. The samples were then transferred in 70% EtOH and mounted in a clearing solution (chloral hydrate:water:-glycerol 8:2:1 w/v/w). The samples were analyzed using transmission light microscopy (Leica DMR, Leica Microsystems, Germany) with differential interference contrast (DIC), using a 20× or 40× dry objective (NA 0.75 and 0.5). Images were acquired using a digital camera (Flexacam C3 Leica Microsystems, Germany).

### Whole mount propidium iodide (PI) staining

Whole-mount staining of ovule primordia was done as described (She and Baroux, 2014). Briefly, Dex-treated flower buds collected at 5 dpi were fixed in 1% formaldehyde, 10% DMSO in PBS-Tween (0.1%), then dissected and embedded in 5% acrylamide pads on microscope slides. Tissue processing included clarification, cell wall digestion and permeabilization before application of 10 μg/ml propidium iodide (PI). Samples were mounted in Vectashield supplemented with PI (Invitrogen).

### Image processing for signal quantification, cell volume measurement, and RHF analysis

Image processing for 3D reconstruction, segmentation, signal, or cell size segmentation was performed with the Imaris 10.2 software (Bitplane, Switzerland). Images presented in the Figures correspond to 3D reconstruction or partial z-projections, using orthogonal or oblique slicers, encompassing the SMCs and surrounding cells.

Cell segmentation for cell size measurements was done as described using ImarisCell, Imaris 10.2 (Mendocilla Sato and Baroux, 2017) based on the Renaissance signal in cell walls (Musielak et al, 2015). Cell volumes of the SMC, the companion cell, and the neighboring cells were exported for analysis.

For quantifying fluorescent reporter signals in nuclei, the later were segmented using Imaris' surface tool, in the semi-automated segmentation mode guided by user-defined manual contours. The sum of voxel intensities (intensity sum) per channel and per nucleus was exported for analysis. If not otherwise indicated, the relative intensity in the SMC was expressed as a ratio between the

intensity sum in the SMC nucleus divided by the average intensity sum of 3–4 nuclei from neighboring L2 nucellar cells.

The relative heterochromatin fraction (RHF) was calculated as the sum of intensity sums in the chromocenters (CC) divided by the intensity sum in the nucleus. Measurements were done on intensity sum projections encompassing the SMC and surrounding nucellus nuclei, processed in Fiji for manual regions-of-interest (ROI) definition.

## 3D protein structure alignment

The 3D predicted protein structures 1VKP for AIH (green) and 2DEW for PADI4 (yellow) were superimposed in COOT (Emsley et al, 2010) and the result was visualized in PyMOL (Chaudhari and Li, 2015).

## Fluorescence recovery after photobleaching (FRAP)

Measurements were done on 1 dpi Dex-induced root tips of 5-day-old seedlings essentially as described (Rosa, 2018) using a confocal laser scanning microscope equipped with a FRAP module (Leica SP8-R, inverted, Leica Microsystems, Germany). Briefly, one sample was prepared at a time, with one root gently mounted (without squashing) in freshly made liquid 0.5x MS. The slide was sealed with transparent nail polish and let to equilibrate for 5–10 min on the microscope stage, with the imaging chamber set at a constant temperature of 20 °C. Bleaching and imaging were done using an HC PL APO 63x/1.20W motCORR CS2 objective lens over a single plane with the pinhole increased up to maximum (5.38 AU) to capture the entire nucleus. Bleaching was performed within a region-of-interest (ROI) of 2-μm diameter, positioned in euchromatin, using 3–5 pulses to reach near full bleach of the signal, using the maximal power and full transmission of the 561 nm laser. Post-bleach images were recorded using low laser intensity for sustainable acquisition, with two sequences: the first sequence of 30 images with 265 ms intervals captures the initial, rapid recovery phase; the second sequence captured the slow recovery phase with 24 images with 10 s intervals. For analyzing fluorescence recovery, images were first corrected for nuclear drifts occurring during acquisition, using a rigid registration approach in Fiji (Schindelin et al, 2012). Fluorescence measurements were done for the bleached ROI, a background ROI of the same size outside the nucleus, and for the nucleus captured manually with contours. The calculation of fluorescence recovery using double normalization was done as described (Rosa et al, 2015) with intensities at each time point expressed relative to the initial intensity (becoming 1, maximum intensity before bleaching) for each image for comparison.

## Fertility analysis

Single inflorescences were induced with either 10 μM Dex or a mock solution as previously described. At 19–21 dpi, the 6th, 7th, 8th silique of each treated plant was collected, and the number of infertile ovules (white, shriveled), aborted seeds (brow, shriveled) and viable seeds (green, plump) were scored.

## RNA in situ hybridization

The antisense RNA probe was designed to target all splicing versions of the *AIH* transcripts (At5g08170) and covers position 441–774 of the coding sequence (see Fig. EV3B). RNA probes were synthesized and labeled using the DIGOXIGENIN (DIG) RNA Labeling Kit (DIG RNA Labeling Mix, Cat. No. 11277073910, Sigma-Aldrich Merck KGaA, Germany) following the manufacturer's instructions. Probe synthesis was performed using the following forward and reverse primers 5'-CGCTTTGGAACGAATTCC-3' and 5'-TAATACGACTCACTA-TAGGGCCGAAAGAGCTTCCACAG-3', respectively. The in vitro transcription reaction was carried out using the T7 RNA polymerase according to the manufacturer's recommendation (Sigma-Aldrich Merck KGaA, Germany). RNA probe preparation and in situ hybridizations were done as described (Dreni et al, 2007). Briefly, single inflorescences (Col_0) were fixed in ice-cold fixative (4% paraformaldehyde (w/v), 1xPBS), embedded in wax using a tissue processor (ASP200, Leica BIOSYSTEMS, Switzerland AG). 6-μm-thick sections were prepared on Superfrost™ Plus Adhesion Microscope Slides (Thermo Fisher Scientific Inc), dewaxed using Histoclear (Sigma-Aldrich Merck KGaA, Germany) and dehydrated using a series of EtOH dilutions. Following Proteinase K digestion, slides were fixed in formaldehyde solution, treated with acetic anhydride and dehydrated using a series of EtOH dilutions. RNA hybridization was performed overnight at 43 °C in 50% formamide, 2xSSC. Post-hybridization steps were carried out with a series of formamide dilutions, before washes and dehydration using an EtOH dilution series. Detection was carried out with an anti-digoxigenin antibody conjugated with alkaline phosphatase (AP), 1:700 (Anti-Digoxigenin-AP, Fab fragments, Sigma-Aldrich Merck KGaA, Germany). The chromogenic reaction was performed overnight using the BCIP alkaline phosphatase substrate and NBT catalyst (BCIP/NBT Color Development Substrate, Promega, Switzerland). Slides were analyzed under transmission light microscopy (Leica DMR, Leica Microsystems, Germany) and images were acquired with a digital camera (Flexacam C3, Leica Microsystems, Germany).

## Measurement of pollen viability

Flower buds at stages 5–7 were treated with Dex as before and flowers at stage 8 to 9 were collected 9 days post induction (9 dpi). Anthers containing mature pollen were dissected, gently squashed onto slides, and fixed in 10% EtOH for 10 min. Samples were then mounted in an Alexander staining solution (10% Ethanol, 0.01% Malachite green, 25% glycerol, 0.05% acid fuchsin, 0.005% Orange G, 4% glacial acetic acid in distilled water (modified after Peterson et al, 2010).

# Data availability

Imaging Dataset used for FRAP, fluorescent signal quantification and cell size measurement: BioStudies S-BIAD2120 (https://doi.org/10.6019/S-BIAD2120, url: https://www.ebi.ac.uk/biostudies/bioimages/studies/S-BIAD2120).

The source data of this paper are collected in the following database record: biostudies:S-SCDT-10_1038-S44318-025-00671-2.

## Peer review information

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

## Acknowledgements

We thank Diana Pazmino, Valeria Gagliardini, Christoph Eichenberger for technical assistance in plant care, digital qPCR, microscopy, respectively, Sara Simonini and Daniela Guthörl (University of Zurich) for advice with RNA in situ hybridization, Stefano Bencivenga (University of Zurich) for advice regarding recalcitrant cloning, Frédérique Pasquer and Ramona Hartman (University of Zurich) for in-house sequencing, Arturo Bolanos and Peter Kopf (University of Zurich) for general lab support, Vimal Rawat (University of Zurich) for help with the expression map of CULLIN genes, the central microscopy facility (ZMB) of the University of Zurich for microscopy support, Mathieu Ingouff

(IRD Montpellier, France) for the *mCHH-YFP* DynaMet line, Mathew Tucker (University of Adelaide, Australia) for the *KNU::nlsYFP* line, Wei Cai Yang (Institute of Genetics and Developmental Biology, Beijing, China) for the *AKV::H2B-YFP* line, Stefan Heckmann (IPK Gatersleben, Germany) for the *ASY1::ASY1-YFP* line, and Daphné Autran (IRD, Montpellier, France) for critical reading of the manuscript and insightful suggestions. The project was supported by the University of Zurich, the Swiss National Science Foundation (grants #310030_185186 and #31003A_149974 to CB, #31003A_179553 to UG, and # 176957 to SB), the European Union's Horizon 2020 research and innovation program under the Marie Sklodowska-Curie (MSC) grant agreement No 847585 (RESPONSE, to CB and SB), and benefitted from networking events of the European Cooperation in Science and Technology (COST) Action # CA16212.

## Author contributions

**Yanru Li**: Formal analysis; Investigation; Visualization; Writing—original draft; Writing—review and editing. **Danli Fei**: Investigation; Visualization; Methodology; Writing—review and editing. **Jasmin Schubert**: Formal analysis; Investigation; Methodology; Writing—review and editing. **Kinga Rutowicz**: Resources; Supervision; Writing—review and editing. **Zuzanna Kaczmarska**: Formal analysis; Visualization; Writing—review and editing. **Alberto Linares**: Investigation; Methodology; Writing—review and editing. **Alejandro Giraldo Fonseca**: Resources; Software; Writing—review and editing. **Sylvain Bischof**: Supervision; Writing—review and editing. **Ueli Grossniklaus**: Resources; Supervision; Funding acquisition; Writing—review and editing. **Célia Baroux**: Conceptualization; Data curation; Formal analysis; Supervision; Funding acquisition; Validation; Investigation; Visualization; Methodology; Writing—original draft; Project administration; Writing—review and editing.

Source data underlying figure panels in this paper may have individual authorship assigned. Where available, figure panel/source data authorship is listed in the following database record: biostudies:S-SCDT-10_1038-S44318-025-00671-2.

## Disclosure and competing interests statement

The authors declare no competing interests.

# Expanded View Figures

**Figure EV1.   H1.1 degradation is controlled by CUL4 and the K89 residue, without altering its stability.**                  ▶

(**A**) Relative Expression of CUL1 (AT4G02570), CUL2 (AT1G02980), CUL3A (AT1G26830), CUL3B (AT1G69670), CUL4 (AT5G46210) in different tissue present in Affymetrix ATH1 GeneCHIP experiments from publicly available datasets (Wuest et al, 2010; Borges et al, 2008; Honys and Twell, 2004; Pina et al, 2005; Schmidt et al, 2011). The dendrogram in the expression heatmap was generated using hierarchical clustering with default options in the heatmap.2 function from the gplots package in R (R Core Team, 2023; Warnes et al, 2024). Pairwise distances were calculated using the Euclidean distance metric, and hierarchical clustering was performed using the complete linkage method. These settings were used to cluster both rows and columns. The Shiny App used to generate the heatmap is available at https://github.com/ AleGirFon/Arabidopsis_heatmap. The table with normalized expression values can be found in Source Data EV1A. Abbreviation: MMC (megaspore mother cell), EmbryoHea (embryo heart stage), EmbryoPreGlo (embryo pre-globular stage). (**B**) Schematic representation of *CULLIN4* (At5g46210) genomic region with indication of the position and sequence of the artificial micro-RNA used in *amiR[CUL4]* lines. (**C**) Western blot (left panel) detection using an anti-CUL4 antibody and Ponceau staining below in *amiRNA [CUL4]* Line #8, treated with Dex and mock, each in two independent replicate extractions. Total proteins were extracted from seedlings at 12 DPI, loaded and immunoblotted as described in the Methods. The expected molecular weight of CULLIN 4 is indicated (~91 KDa). Right: quantification of the relative CUL4 band intensity (see Methods) in three independent lines (4, #7 and #8), each in two replicate protein extraction. Blots used for quantifications in Source Data EV1C. *P* value: Mann–Whitney U test. Boxplots: Center lines indicate medians; boxes span the interquartile range (25th–75th percentiles); whiskers extend 1.5× the interquartile range beyond the box limits, as computed in R. (**D**) *CUL4* knockdown as in the Dex inducible *amiRNA[CUL4]* line recapitulates a known f*usca* phenotype (Chen et al, 2006). Seedlings are 11DAG-old and were grown on ½ MS supplemented with 10 µM DEX (upper panel) or a MOCK solution (bottom panel). Whit arrows indicate brownish pigmentation in the cotyledons, representing the f*usca* phenotype, Scale bar = 5 mm. (**E**) Screen shot from the Plant PTM Viewer webserver. The Plant PTM Viewer shows six putative ubiquitination sites on Arabidopsis H1.1 (Willems et al, 2019). (**F**) Representative images of H1.1-RFP, H1.1^6xGC^-RFP in a wild-type background and co-expressed with H1.1-GFP under its native promoter (She et al, 2013) showing persistence of H1.1^6xGC^-RFP and depletion of H1.1-GFP in the SMC (dashed lines) of ovule primordia 5 dpi; Dotted border outlines the SMC. Pie charts show replicate measurements of persistence vs depletion categories in independent lines. n: number of scored ovule primordia. *P* values, Fisher exact test. (**G**) Fluorescent Recovery After Photobleaching experiments measuring the mobility of the different H1.1 variants as indicated in seedling roots and Boxplot showing the recovery rate at 30 s, 60s- and 4-min. *n*: number of analyzed nuclei; error bar: standard error to mean. *P* values, Mann–Whitney U test. Boxplots: Center lines indicate medians; boxes span the interquartile range (25th–75th percentiles); whiskers extend 1.5× the interquartile range beyond the box limits, as computed in R. (**H**) Alignment of selected Arabidopsis and mouse H1 variants (AtH1.1: AT1G06760 H1.1; AtH1.2: AT2G30620 H1.2, Mouse H1.0: P10922, Mouse H1.1: P43275, Mouse H1.2: P15864) cropped around the globular domain (yellow) showing the position of the conserved Lysin residue (K89 in AtH1.1) and the three α helices as indicated. Right: representation of AtH1.1 3D folding of the globular domain, indicating the position of K89. (**I**) Replicate measurements of H1.1^K89R^ persistence vs depletion in the SMC of three independent lines induced as described in the main text and compared to a control line. Dotted border outlines the SMC. *P* value: Fisher exact test. (**J**) 3D projection showing H1.1^6GC^-RFP and H1.1^R57K^-RFP persistence in both euchromatin and heterochromatin. Top panel: representation of the image processing used to isolate nuclei in silico for projections (left, whole primordium counterstained with Renaissance; middle, primordium after segmentation and masking of the SMC nucleus (magenta) and several nucellus nuclei (cyan); right: nuclei projection only). Middle and bottom panels: three representative images for the H1.1 variants as indicated. Scale bar: 5 µm. (**K**) Sequence alignment of the forward (F) and reverse (R) sequencing fragments from the transgenes of 12 independent transgenic lines (DLF15-26) covering the *H1.1* transgene region revealing the eleven engineered A to G mutations (red), enabling K to R substitutions, including at residue K89. Images below show low H1.1^11xG^-RFP signals in ovule primordia in comparison to H1.1^6xGC^-RFP. Dotted border outlines the SMC. See also Source data Fig. EV1 and Table EV1. Source data are available online for this figure.

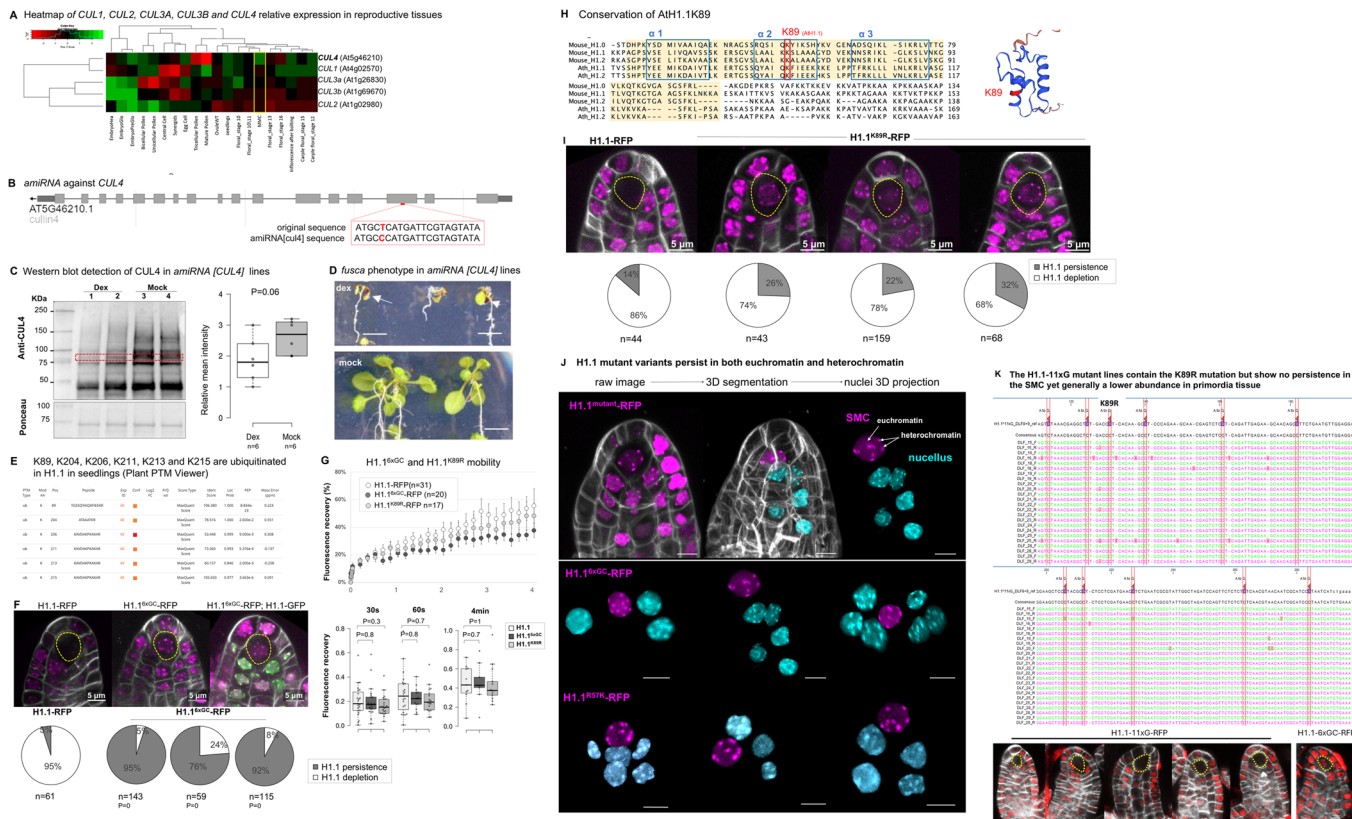

**A** Heatmap of CUL1, CUL2, CUL3A, CUL3B and CUL4 relative expression in reproductive tissues

**B** amiRNA against CUL4

**C** Western blot detection of CUL4 in amiRNA [CUL4] lines

**D** fusca phenotype in amiRNA [CUL4] lines

**E** K89, K204, K206, K211, K213 and K215 are ubiquitinated in H1.1 in seedlings (Plant PTM Viewer)

**F** H1.1-RFP, H1.1⁶ˣGC-RFP, H1.1⁶ˣGC-RFP; H1.1-GFP

**G** H1.1⁶ˣGC and H1.1^K89R mobility

**H** Conservation of AtH1.1K89

**I** H1.1-RFP, H1.1^K89R-RFP

**J** H1.1 mutant variants persist in both euchromatin and heterochromatin

**K** The H1.1-11xG mutant lines contain the K89R mutation but show no persistence in the SMC yet generally a lower abundance in primordia tissue

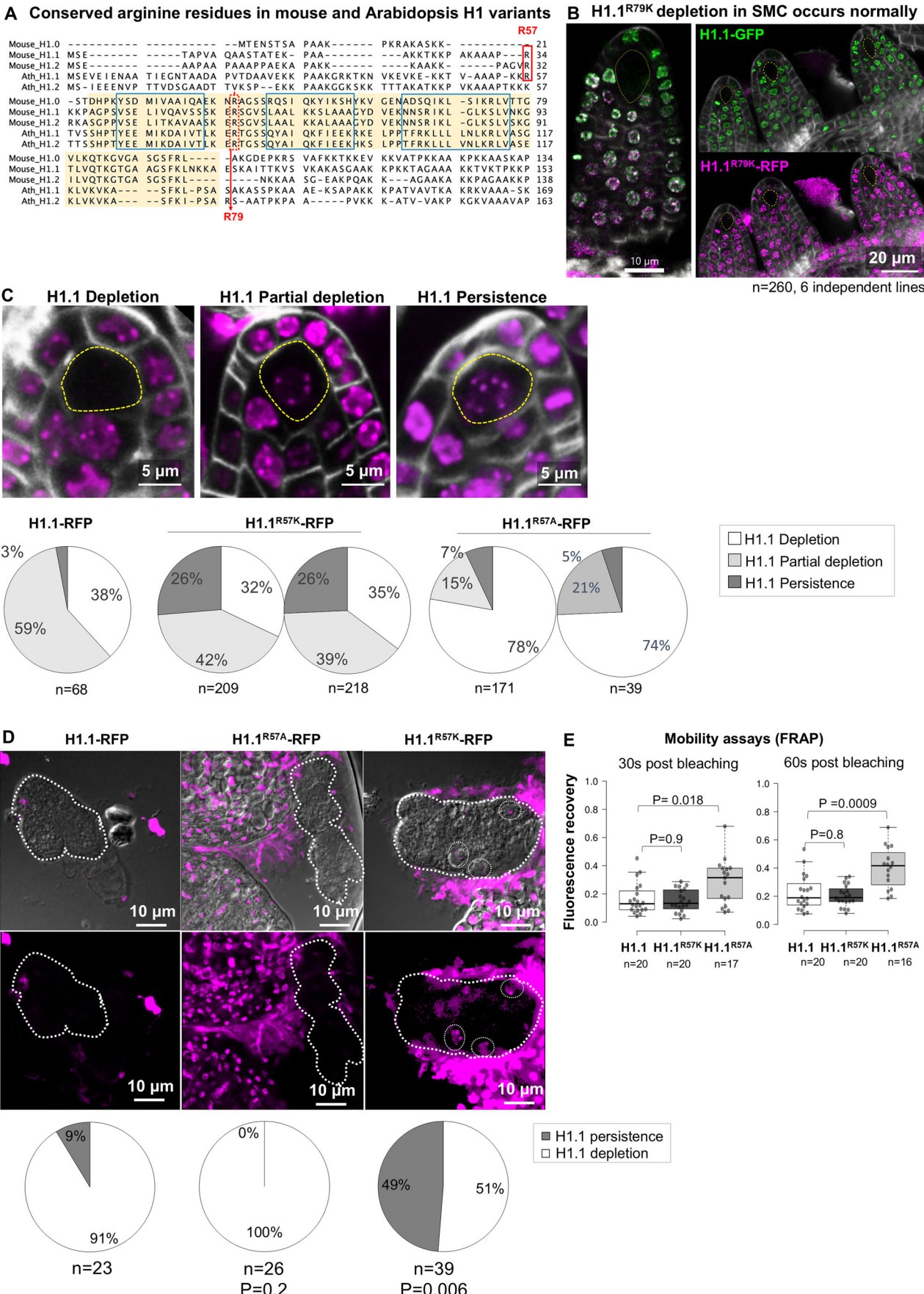

**A** Conserved arginine residues in mouse and Arabidopsis H1 variants

**B** H1.1^R79K depletion in SMC occurs normally

n=260, 6 independent lines

**C**  H1.1 Depletion     H1.1 Partial depletion     H1.1 Persistence

☐ H1.1 Depletion
░ H1.1 Partial depletion
▓ H1.1 Persistence

H1.1-RFP
n=68

H1.1^R57K-RFP
n=209    n=218

H1.1^R57A-RFP
n=171    n=39

**D**  H1.1-RFP     H1.1^R57A-RFP     H1.1^R57K-RFP

**E**  Mobility assays (FRAP)

▓ H1.1 persistence
☐ H1.1 depletion

n=23     n=26     n=39
          P=0.2    P=0.006

◀  **Figure EV2.  H1.1 depletion in the SMC is controlled by the R57 residue (related to Fig. 2).**

(A) Alignment of selected Arabidopsis and Mouse variants as those in Fig. EV1 (globular domain highlighted in yellow) showing the conservation of arginine residues (red boxes). The R57 residue from AtH1.1 studied in this work is positioned in the N-tail just before the globular domain. It is conserved in the Mouse H1.1 and H1.2 variants but not in the Arabidopsis H1.2 variant. The arginine shown by Christophorou and colleagues to be citrullinated is R54 in the Mouse H1.2 variant corresponds to R79 in AtH1.1 and AtH1.2. (B) H1.1$^{R79K}$-RFP (magenta) is normally depleted in SMCs (dotted outline),  as is the native H1.1-GFP variant (green). Ovule primordia at 5 dpi were counterstained with Renaissance (gray). (C)  Representative images showing either full H1.1 depletion, partial depletion or persistence in the SMC (dotted outline) of ovule primordia stage 2-I at 5 dpi (partial confocal projections) as used for scoring shown in the pie charts below, for the H1.1-RFP control line and two independent lines expressing the mutant variants as indicated. *n*, number of primordia scored. The images illustrating H1.1 Depletion and H1.1 Persistence are reused from Fig. 2B for facilitating comparisons. See *P* values from a chi2 contingency test comparing the distribution of the three categories Table EV1. (D) In premeiotic sporangia from flowers at 3 dpi, H1.1-RFP and H1.1$^{R57A}$-RFP are depleted in male SMC (pollen mother cells, dotted outline), but H1.1$^{R57K}$-RFP shows residual signal (magenta). Confocal images show overlays of RFP signal (magenta) with transmission light images with differential interference contrast (DIC); pie charts quantify the observations. *n*, number of sporangia. Dotted border outlines Pollen Mother Cells (PMC). (E) Fluorescence recovery rate from FRAP experiments shown Fig. 2, at 30 s and 60 s post bleaching. *n*: number of analyzed nuclei. Boxplots: Center lines indicate medians; boxes span the interquartile range (25th–75th percentiles); whiskers extend 1.5× the interquartile range beyond the box limits, as computed in R. *P* value, Mann–Whitney U test. See also Source data Fig. EV2 and Table EV1. Source data are available online for this figure.

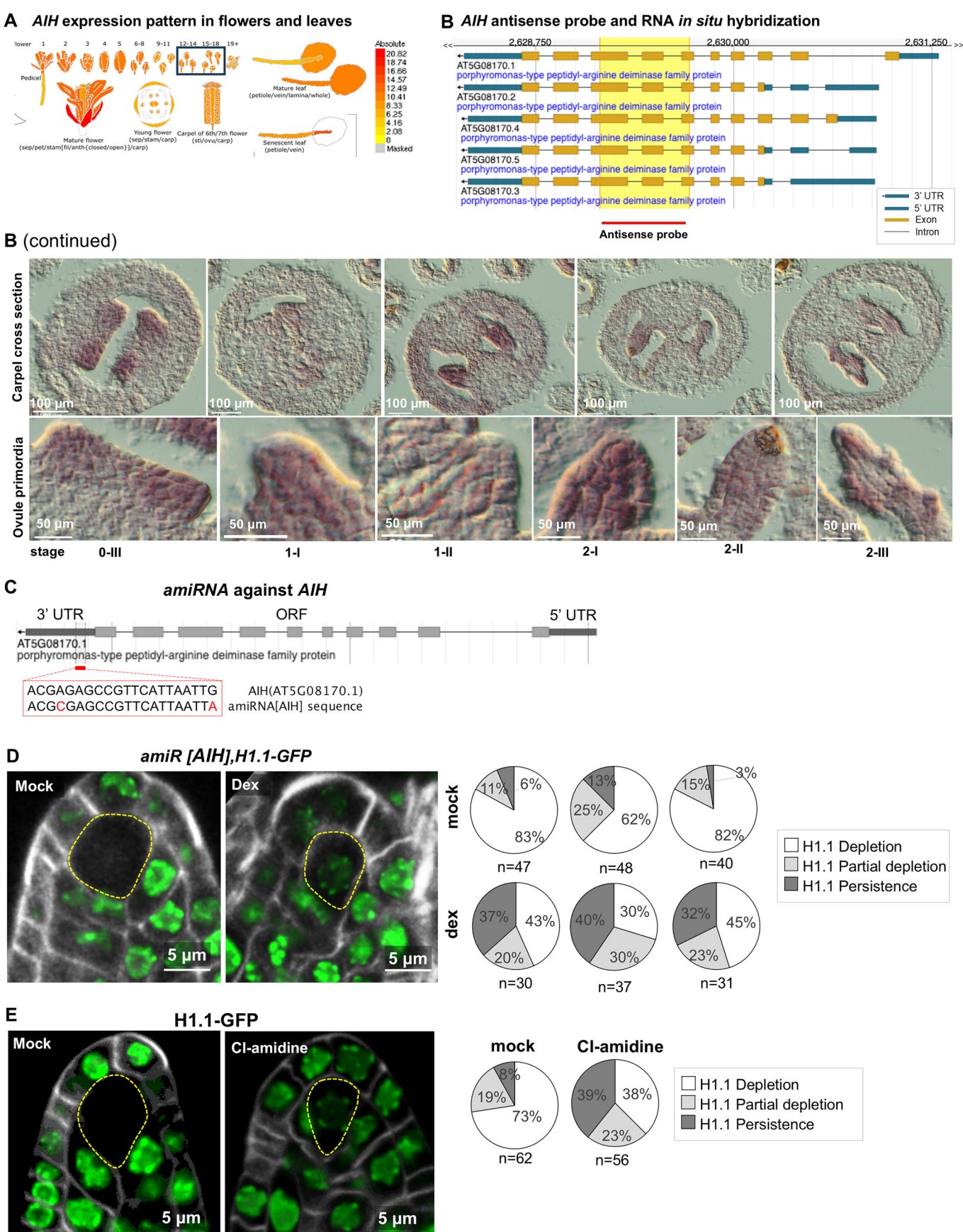

**A**  *AlH* expression pattern in flowers and leaves

**B**  *AlH* antisense probe and RNA *in situ* hybridization

**B** (continued)

Carpel cross section

Ovule primordia

stage    0-III          1-I          1-II          2-I          2-II          2-III

**C**    *amiRNA* against *AlH*

3' UTR          ORF          5' UTR

AT5G08170.1
porphyromonas-type peptidyl-arginine deiminase family protein

ACGAGAGCCGTTCATTAATTG          AlH(AT5G08170.1)
ACGCGAGCCGTTCATTAATTA          amiRNA[AlH] sequence

**D**    *amiR [AlH],H1.1-GFP*

Mock          Dex

mock

dex

H1.1 Depletion
H1.1 Partial depletion
H1.1 Persistence

**E**    H1.1-GFP

Mock          Cl-amidine

mock          Cl-amidine

H1.1 Depletion
H1.1 Partial depletion
H1.1 Persistence

Figure EV3.   The AIH citrullinase mediates H1.1 depletion in SMC (related to Fig. 3).

(A) Selected view of *AIH* expression generated by the ePlant Browser (bar.utoronto.ca/eplant/) showing middle-to-strong expression in young flower buds (box). (B) Schematic representation of the five splice variants of *AIH* (TAIR resource, arabidopsis.org) and the position of the probe used for RNA in situ hybridization shown below. Top panel: cross section through carpels, Bottom panel: Magnified views of the top panel, illustrating ovule primordia from the corresponding carpels at the indicated developmental stages (0-III to 2-III) following the nomenclature (Hernandez-Lagana et al, 2021). The third image in the top panel and the stage 2-I image in the bottom panel are reused from Fig. 3B to facilitate comparisons. (C) Position and sequence of the amiRNA used for downregulating *AIH*. (D, E) Representative images and scoring in three independant lines (piecharts) of primordia showing H1.1-GFP in the SMC following *AIH* downregulation using the inducible *amiR[AIH]* (D) or AIH inhibition using Cl-amidine (E). Top panels: images of mock and Dex (D), or mock and Cl-amidine (E) treated ovule primordia at 5 dpi expressing H1.1-GFP under its native promoter (She et al, 2013) and the inducible *amiR[AIH]* as described in the main text. Dotted contours outline the SMC. Pie charts: replicate scoring of depletion/partial depletion/persistence pattern of H1.1-GFP in the SMC, in three independent lines. *n*, number of ovule primordia scored, *P* values testing the distribution of all three categories: see Table EV1. Confocal images in (D) and (E) are reused from Figs. 3C and 3D, respectively, for facilitating comparisons. See also Source data Fig. EV3 and Table EV1. Source data are available online for this figure.

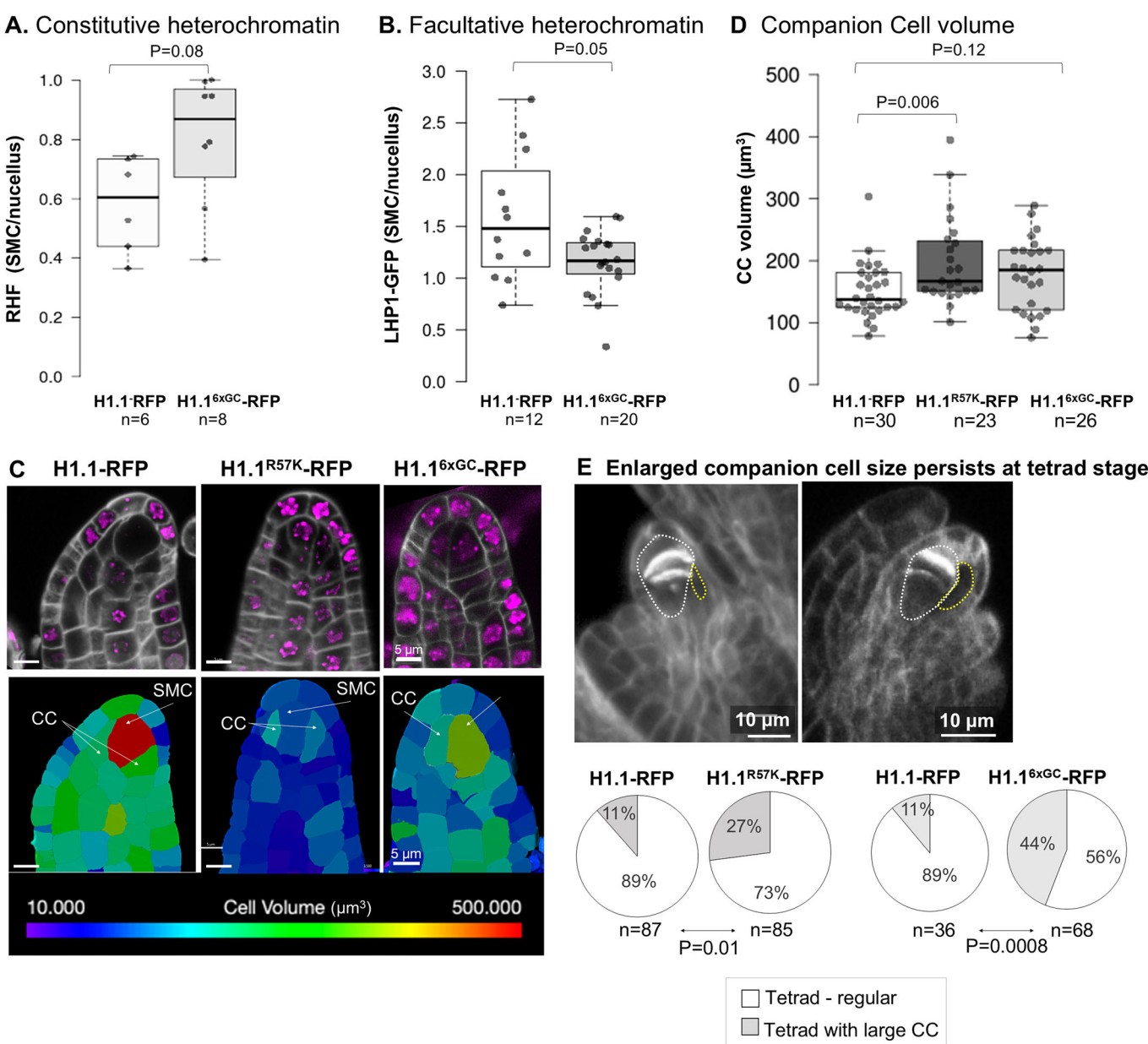

**Figure EV4. Effect of H1.1 persistence in the SMC on chromatin, SMC maturation and meiosis (related to Fig. 4).**

(A, B) Relative Heterochromatin Fraction (RHF, A) and LHP1-GFP levels (B) in the SMC relative to surrounding nucellar cells, in the H1.1-RFP control line and H1.1^6xGC-RFP mutant line. (C) Representative images and corresponding cell-based segmentation (below) of ovule primordia at 5 dpi expressing the control of mutant H1.1 variants as indicated, to measure the SMC and CC volumes as plotted in Fig. 4 and Panel D here. The segmented image for H1.1-RFP is reused from Fig. 4D to facilitate comparisons. (D) Volume of the companion cells (CC) of ovule primordia at 5 dpi expressing the control or mutant H1.1 variants as indicated. (E) Representative images of ovules at the tetrad stage showing a tetrad (white dotted line) and a neighboring, narrow or enlarged CC (left and right, respectively, yellow dotted line). Pie charts showing scoring of the respective classes in ovule primordia at 6 dpi, induced for the expression of the control or mutant H1.1 variants as indicated. *n*, number of ovule primordia scored. *P* values, Mann–Whitney U test (A, B, D) or Fisher exact test (C). Boxplots: Center lines indicate medians; boxes span the interquartile range (25th–75th percentiles); whiskers extend 1.5× the interquartile range beyond the box limits, as computed in R. See also Source data Fig. EV4 and Table EV1. Source data are available online for this figure.

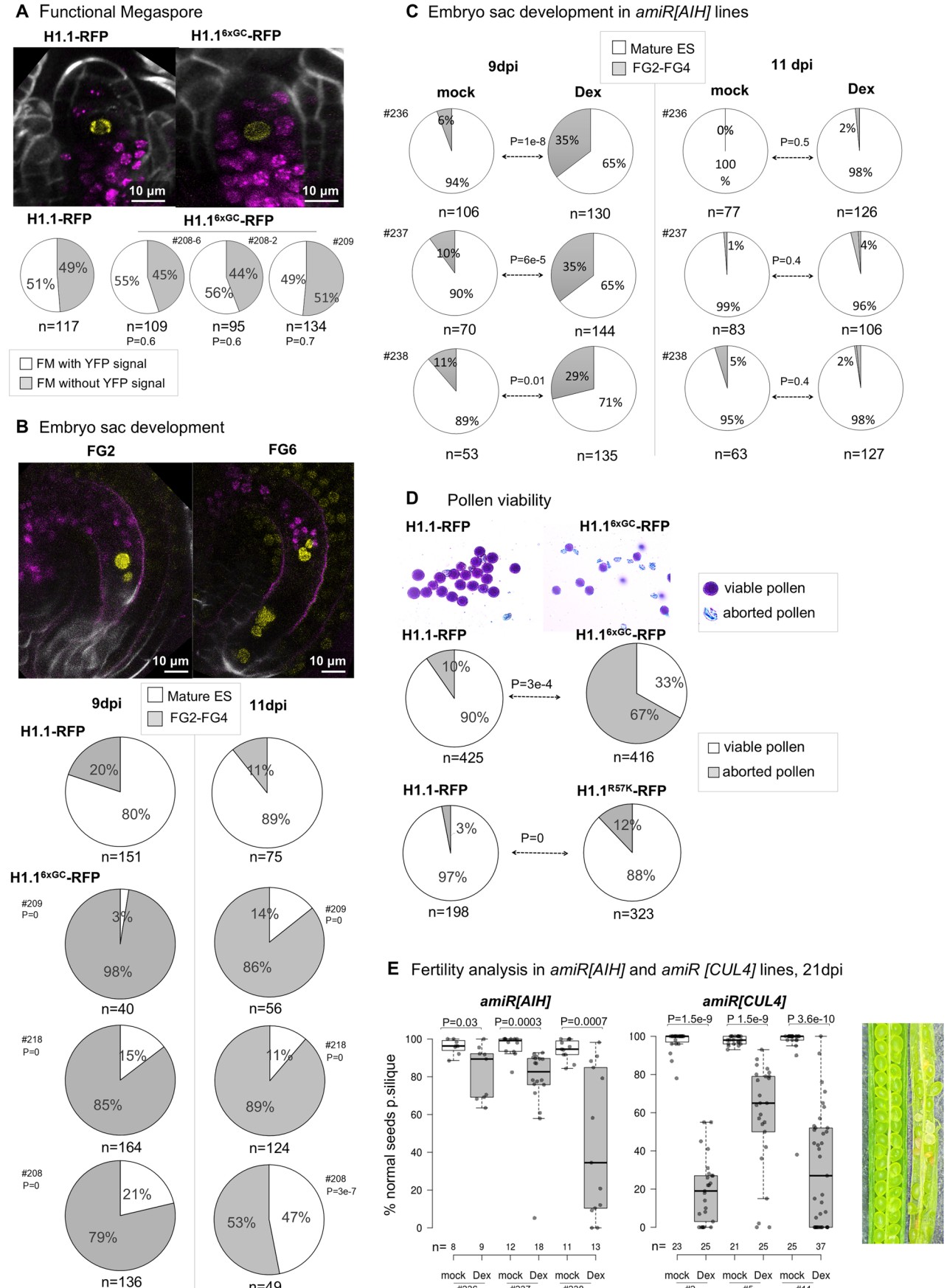

**A** Functional Megaspore

**B** Embryo sac development

**C** Embryo sac development in *amiR[AIH]* lines

**D** Pollen viability

**E** Fertility analysis in *amiR[AIH]* and *amiR [CUL4]* lines, 21dpi

◀

**Figure EV5.   Impact of H1.1 persistence in the SMC on embryo sac development and fertility (related to Fig. 5).**

(A) Representative images and scoring of ovule primordia at 7 dpi expressing *AKV::H2B-YFP* in the functional megaspore in the control or H1.1[6xGC]-RFP line (3 independent lines). The image for H1.1-RFP is reused from Fig. 5A for facilitating comparisons. (B) Representative images and scoring of ovules at 9 dpi and 11 dpi (two days after emasculation at 9 dpi) expressing *AKV::H2B-YFP* in the embryo sac (stages FG2 and FG6 are shown) in the control line and scoring of these classes in the control and H1.1[6xGC]-RFP lines (3 independent lines). (C) Scoring of FG2-FG4 and FG6 embryo sacs identified by clearing, in ovules at 9 dpi and 11 dpi (two days after emasculation at 9 dpi) following the induction of *amiR[AIH]*. (D) Assessment of pollen viability by Alexander staining in the control and mutant lines as indicated, by scoring anthers at 9 dpi. (A–E) *P* values, Fisher exact test. (E) Fertility analysis by quantifying the % of normal (green, plump) seeds per silique at 21 dpi after mock or dex treatment, in mutant lines as indicated, in three independent lines. *n*, number of siliques scored. *P* values from a Mann–Whitney U test. Boxplots: Center lines indicate medians; boxes span the interquartile range (25th–75th percentiles); whiskers extend 1.5× the interquartile range beyond the box limits, as computed in R. See also Source data Fig. EV5 and Table EV1. Source data are available online for this figure.

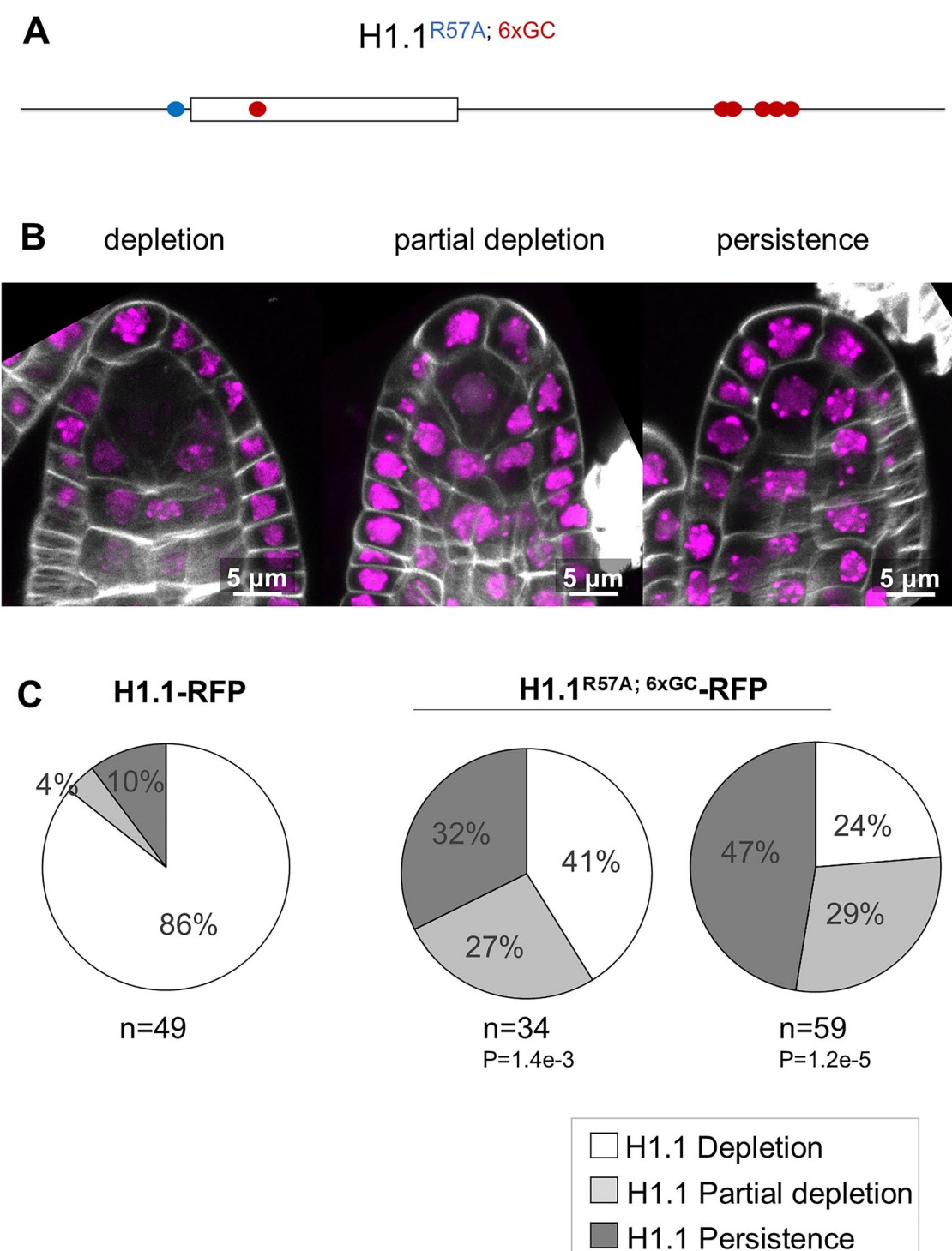

**Figure EV6.    An H1.1 variant combining the R57A and K89R mutations shows resistance to degradation.**

(A) Schematic representation of the H1.1^R57A;6xGC mutant variant showing the mutated R57A residue (blue) and the 6 K-to-R substitutions (red) among which K89R in the globular domain (box). (B) Representative images of the depletion, partial depletion and persistence phenotype as scored in (C). Note the diffused pattern in the 'partial' category indicating increased dissociation from heterochromatin suggested to be a result of R57A. (C) Scoring in one control line and two independent double mutant lines as indicated. *n*, number of ovule primordia scored at 5 dpi. *P* values, Chi-square contingency test of the three categories (depletion, partial depletion, persistence) distribution. See also Source data Fig. EV6 and Table EV1. Source data are available online for this figure.

