## [Peer Review File · The EMBO Journal]

Pre-meiotic H1.1 degradation is essential for Arabidopsis gametogenesis

Yanru Li, Danli Fei, Jasmin Schubert, Kinga Rutowicz, Zuzanna Kaczmarek, Alberto Linares, Alejandro Fonseca, Sylvain Bischof, Ueli Grossniklaus, and Celia Baroux

Corresponding author(s): Celia Baroux (cbaroux@botinst.uzh.ch)

Review Timeline:

Submission Date:	24th Jun 25
Editorial Decision:	14th Aug 25
Revision Received:	6th Sep 25
Editorial Decision:	23rd Oct 25
Revision Received:	30th Oct 25
Accepted:	25th Nov 25

Editor: Cornelius Schneider

Transaction Report:

Dear Dr. Baroux,

Thank you for submitting your manuscript for consideration by the EMBO Journal. It has now been seen by three referees whose comments are shown below.

Given the referees' positive recommendations, I would like to invite you to submit a revised version of the manuscript, addressing the comments of all three reviewers. I should add that it is EMBO Journal policy to allow only a single round of revision, and acceptance of your manuscript will therefore depend on the completeness of your responses in this revised version.

Thank you for the opportunity to consider your work for publication. I look forward to your revision. Please do not hesitate to contact me if you have any question regarding the revisions. I am also happy to discuss specific revision experiments by e-mail or videoconferencing.

Yours sincerely,

Cornelius Schneider, PhD
Editor
The EMBO Journal
c.schneider@embojournal.org

Please remember: Digital image enhancement is acceptable practice, as long as it accurately represents the original data and

conforms to community standards. If a figure has been subjected to significant electronic manipulation, this must be noted in the figure legend or in the 'Materials and Methods' section. The editors reserve the right to request original versions of figures and the original images that were used to assemble the figure.

We realize that it is difficult to revise to a specific deadline. In the interest of protecting the conceptual advance provided by the work, we recommend a revision within 3 months (12th Nov 2025). Please discuss the revision progress ahead of this time with the editor if you require more time to complete the revisions. Use the link below to submit your revision:

Referee #1:

This study provides compelling evidence for the essential role of H1.1 degradation prior to gametogenesis in Arabidopsis and proposes a citrullination-ubiquitination cascade involving AIH and CUL4 as a mechanistic basis for chromatin remodeling during reproductive cell fate transition. The findings are conceptually significant and point to a potentially conserved mechanism shared with animal systems. The proposed two-step model-AIH-mediated charge neutralization followed by CUL4-mediated degradation-is well supported by biochemical, genetic, and imaging evidence.

To further strengthen the manuscript, I offer the following suggestions:

1. AIH is proposed to act as a citrullinase based on structural homology and Cl-amidine sensitivity. However, AIH is also known to function in polyamine metabolism. It would be helpful for the authors to discuss whether AIH is likely to act specifically on H1.1 in vivo, or whether it may have additional nuclear or non-histone substrates that could influence the observed phenotypes.
2. Given that both AIH and CUL4 are expressed broadly within the ovule primordium, it remains unclear how H1.1 degradation is restricted to the SMC. Exploring potential mechanisms such as spatially restricted subcellular localization, post-translational activation, or cofactor dependence (e.g., calcium ions as in PAD4-mediated citrullination in mammals) could further clarify how cell specificity is achieved.
3. The study distinguishes the developmental effects of H1.1R57K and H1.1 6GC mutants, but additional quantification-such as the number, size, and positioning of nuclei during embryo sac development-would enhance the interpretation of the observed delays or arrests.
4. To better distinguish whether the embryo sac phenotypes represent true developmental arrest or delayed progression, it would be valuable to assess the expression of stage-specific marker genes (e.g., via in situ hybridization or fluorescent reporters) in the mutant backgrounds. This would help link chromatin state with transcriptional dynamics.
5. As the study relies heavily on Dex-inducible constructs and artificial miRNAs, the authors should include controls to address potential off-target or Dex-related effects. For example, assessing embryo sac development in lines expressing unrelated amiRNAs or Dex-inducible reporters not targeting H1.1 would help establish the specificity of the observed phenotypes.

Referee #2:

This study investigates the role of linker histone H1 eviction during differentiation of spore mother cells (SMCs) in Arabidopsis thaliana. The authors propose that H1 eviction is controlled by two post-translational modifications: (i) ubiquitination at K89, mediated by the CULLIN4 E3 ligase, which targets H1.1 for degradation via the proteasome; and (ii) citrullination at R57, which destabilizes H1.1 by weakening its binding to chromatin. They propose that a potential plant citrullinase, AIH, catalyzes citrullination at R57. Downregulation or inhibition of AIH (by Cl-amidine) partially impairs H1.1 depletion. Furthermore, the authors find that H1.1 depletion is crucial for gametogenesis but not for meiosis. Based on these findings, they propose a two-step eviction model: in step 1, citrullination destabilizes H1.1 and increases the unbound pool of H1; in step 2, the unbound H1.1 is ubiquitinated by CUL4 and subsequently degraded.

While the research question is relevant and interesting, I have several concerns regarding the data and the conclusions that would need to be addressed.

1. AIH and CUL4 are broadly expressed, yet H1.1 eviction is specific to the SMC. What confers this specificity?
2. The authors propose that a single H1 arginine residue is the primary target for citrullination. However, mutating R57 to K results in only 26% of SMCs with persistent H1.1, a relatively minor effect compared to the control (11% persistent). If citrullination of H1.1 precedes CUL4-mediated degradation, one would expect that blocking citrullination would have a similar consequence to blocking CUL4-yet this is not the case. It would therefore be important to also test the effect of the R79K mutation, which, as the authors note, is the corresponding citrullinated residue in mouse H1. The statement "the R57K 18 mutation compromised H1.1 depletion in a large fraction of ovule primordia" should be toned down and adjusted to the findings.
3. Figure 1C: The 11xG mutants are not described in the Results section. It is difficult to understand how the K89R mutation has

an effect, while mutation of 11 lysine residues, including K89, does not. These data are mentioned in the discussion, but should be addressed in the Results section as well.

4. Figure 5C: The R57K line exhibits a high level of seed abortion, which appears inconsistent with its relatively minor effect on female gametophyte development. Furthermore, to show that the 6xGC line indeed impairs female gametophyte function, the authors should pollinate this line with wild-type pollen and then score seed development.

5. Based on the claims of the authors, mutations in CUL4 should lead to a female gametophytic effect. However, this was not tested by the authors; instead they tested seed abortion upon self-pollination, which does not allow to conclude whether the female gametophyte is functionally impaired. They should pollinate their lines with wild-type pollen to test whether this impairs seed development. It would add to this manuscript if the authors would include the hypomorphic *cul4-1* allele (Bernhardt et al., 2006), which should recapitulate the findings reported using the *amiR*-lines.

6. Please correct: Female gametophyte development does not involve endomitosis (DNA replication without subsequent mitosis), and the female gametophyte before cellularization is a coenocyte, not a syncytium (which is a structure developing upon cell fusion).

Referee #3:

This work is aimed at understanding the loss of histone H1 in SMC maturation, a process already reported earlier by the same laboratory. Now authors develop an elegant strategy to establish that H1 is degraded via the proteasome which is analogous to the H1 modification and degradation reported in the mouse. They found that this is dependent on CUL4 and identify two amino acid residues necessary for efficient degradation, K79 and R57. In addition to ubiquitination of K79 that marks H1 for proteolysis, citrullination of R57 is necessary for H1.1 depletion. This step is carried out by AIH, an arginine deiminase. They also try to delineate the functional relevance of these modifications and found that H1 persistence does not impair SMC maturation or meiosis but are important for SMC growth and later establishment of the embryo sac and seed fertility.

The work is experimentally correct and I do not have major concerns. However, addressing my comments below may improve the quality and impact of the study.

1. Section describing the screen for potential Ubiquitination sites. The rationale to select the residues to be mutated seems to be based on homology studies. Is this the case? It could be worth explaining in more detail the process to select a single amino acid residue, e.g., K79, for further analysis.
2. Impairment of H1 depletion only moderately affects SMC growth but not expression of specific markers. Given the importance of chromatin remodeling this seems unexpected and may deserve a more detailed discussion.
3. H1 depletion appears to be important for the embryo sac development. Since this step requires multiple divisions of the haploid nuclei and H1 contains several putative CDK phosphorylation sites it could be relevant exploring a possible relationship with H1 modifications. I understand that this is beyond the scope of the present study but I wonder if roscovitine could be used to answer this.
4. It may be worth including in the discussion a recent article (• DOI: 10.1038/s41586-025-08835-0) on H1 deamidation and its role in chromatin relaxation.
5. The mechanism proposed here is quite sophisticated. Do the authors consider that it is applicable exclusively to SMC or may operate also at other stages of plant life or other organs? It is known that a different H1 variant (H1.3) is expressed in response to abiotic stress and likely replaces H1.1 and H1.2. Could a similar mechanism apply in seedlings in response to abiotic stress? I suggest including this in the discussion.
6. There are a few typos. Page 9, line 24, amin... ; Page 16, line 33, Too ... Please check throughout.

Referee #1:

This study provides compelling evidence for the essential role of H1.1 degradation prior to gametogenesis in Arabidopsis and proposes a citrullination-ubiquitination cascade involving AIH and CUL4 as a mechanistic basis for chromatin remodeling during reproductive cell fate transition. The findings are conceptually significant and point to a potentially conserved mechanism shared with animal systems. The proposed two-step model-AIH-mediated charge neutralization followed by CUL4-mediated degradation-is well supported by biochemical, genetic, and imaging evidence. To further strengthen the manuscript, I offer the following suggestions:

1. AIH is proposed to act as a citrullinase based on structural homology and Cl-amidine sensitivity. However, AIH is also known to function in polyamine metabolism. It would be helpful for the authors to discuss whether AIH is likely to act specifically on H1.1 in vivo, or whether it may have additional nuclear or non-histone substrates that could influence the observed phenotypes.

>>We thank the reviewer for suggesting extending the discussion on this question. We **added** the following sentences “Although a direct role of AIH in regulating H1.1 stability via arginine citrullination is the most straightforward explanation based on our findings, we cannot rule out the possibility that AIH contributes to H1.1 depletion indirectly through its role in polyamine metabolism. Polyamines have emerged as important regulators of both biotic and abiotic stress responses in plants, as well as various developmental processes, including flowering, seed development, and seedling establishment (Blázquez, 2024). Notably, the *aih* loss-of-function mutant is embryo-lethal, underscoring its essential role in development (Blázquez, 2024). Also, supporting the idea that polyamines can influence chromatin structure, spermidine deficiency—induced in the water fern *Marsilea vestita*—has been shown to impair chromatin compaction and nuclear elongation in male gametophytes (Deeb et al, 2010).”

2. Given that both AIH and CUL4 are expressed broadly within the ovule primordium, it remains unclear how H1.1 degradation is restricted to the SMC. Exploring potential mechanisms such as spatially restricted subcellular localization, post-translational activation, or cofactor dependence (e.g., calcium ions as in PAD4-mediated citrullination in mammals) could further clarify how cell specificity is achieved.

>> We thank the reviewer for this insightful comment. As discussed in the manuscript, we already consider potential mechanisms that could confer spatial specificity to AIH activity within the SMC, page 22: “A key question arising from our findings is the cell specificity of H1.1 degradation. Both AIH and CUL4 are broadly expressed in ovule primordia, a pattern that does not explain the observed cell-specific degradation. One possibility is that these enzymes might be catalytically regulated by a cofactor conferring cell specificity. For example, mammalian PADI4 activity is influenced by Ca²⁺ availability (Arita et al, 2004). In an alternative model, increased H1.1 dissociation followed by degradation may be the default process in all cells of the primordium but low or stalled transcription and/or translation in the SMC could prevent replenishment of the H1.1 pool.”

We fully agree that uncovering the mechanism underlying SMC-specificity in the action of AIH and CUL4 would be of significant interest. However, pursuing such an experimental direction—such as targeted manipulation of Ca²⁺ levels in the SMC or identifying an SMC-specific post-translational modification (PTM) of AIH—lies beyond the scope of the current study. Notably, mass spectrometry-based techniques are not yet applicable to this rare cell type. In this work, we focus on reporting and functionally characterizing a novel pathway regulating H1 dynamics in germline precursor cells. Nonetheless, we acknowledge the importance of this question and have expanded the discussion accordingly.

“A key question arising from our findings is the cell specificity of H1.1 degradation. Both AIH and CUL4 are broadly expressed in ovule primordia, a pattern that does not explain the observed cell-specific degradation. One possibility is that these enzymes might be catalytically regulated by a cofactor conferring cell specificity. For example, mammalian PADI4 activity is influenced by Ca²⁺ availability (Arita *et al*, 2004). Although calcium showed undetectable levels in male SMCs in *Nicotiana tabacum* and *Torenia fournieri* (Ge *et al*, 2008), no studies have yet been conducted on calcium availability in ovule primordia or in SMCs. Also, considering the intricate interplay between calcium, reactive oxygen species, and the cellular availability of free oxygen (Görlach *et al*, 2015), as well as the recent discovery that hypoxia regulates protein degradation and stem cell fate in the shoot apical meristem (Weits *et al*, 2019) and can induce meiotic fate acquisition in maize (Kelliher & Walbot, 2012), exploring the role of hypoxia in SMC differentiation—particularly in relation to chromatin remodeling—presents an intriguing avenue for future research, which will require solving challenges in experimental approaches. A second possibility is that of an SMC-specific AIH isoform encoded by one of the distinct splice variants (**Figure EV3B**). Finally, an interesting alternative possibility contrasts with the notion of specific activity of AIH and CUL4. Instead, it posits H1.1 dissociation - degradation as the default process in all cells of the primordium corresponding to the broad domain of AIH and CUL4 expression, but an SMC-specific lack of transcriptional and/or translational compensation prevents the replenishment of the H1.1 pool in these cells.”

We hope the reviewer finds this clarification satisfactory

3. The study distinguishes the developmental effects of H1.1R57K and H1.1 6GC mutants, but additional quantification—such as the number, size, and positioning of nuclei during embryo sac development—would enhance the interpretation of the observed delays or arrests.

>> the conclusion of embryo sac delay or arrest is already based on the scoring on the number of nuclei which is used to determine the stages FG1 until FG7. We clarify the text by **modifying** the relevant sentence: “We then analysed the embryo sacs formed by these functional megaspores, using the number of nuclei as a stage criterion. Whereas the control showed a majority of mature (FG6) embryo sacs in flower buds just before anthesis a significant proportion of embryo sacs with earlier stages (FG2-FG4) were observed when either H1.1 mutant variant was induced”. To differentiate between arrest and delay, we conducted an experiment **presented Figure 5B** in which the embryo sac stage was scored two days after emasculation, a classic method for assessing this. As expected, the results revealed that expression of the H1.1-R57K variant led to a delay in

embryo sac development, while expression of the H1.1-6GC variant resulted in complete arrest. We interpret this difference based on our model, where the H1.1-R57K variant exhibits a delayed degradation dynamic, whereas the H1.1-6GC variant is protected from degradation.

4. To better distinguish whether the embryo sac phenotypes represent true developmental arrest or delayed progression, it would be valuable to assess the expression of stage-specific marker genes (e.g., via in situ hybridization or fluorescent reporters) in the mutant backgrounds. This would help link chromatin state with transcriptional dynamics.

>> As explained above, to differentiate between arrest and delay, we conducted an experiment **presented Figure 5B** in which the embryo sac stage was scored two days after emasculation, a classic method for assessing this. As expected, the results revealed that expression of the H1.1-R57K variant led to a delay in embryo sac development, while expression of the H1.1-6GC variant resulted in complete arrest. We interpret this difference based on our model, where the H1.1-R57K variant exhibits a delayed degradation dynamic, whereas the H1.1-6GC variant is protected from degradation.

To our knowledge, no stage-specific markers currently differentiate FG2, FG4, and FG6 stages. We already use the AKV::H2B-YFP marker, which is specifically active from FG1 onward. An egg cell-specific marker would not provide information about earlier stages. Instead, we plan to perform transcriptional profiling of embryo sacs derived from SMCs in which H1 depletion was inhibited. However, this requires laser capture of FG-stage cells and is therefore beyond the scope of the present study.

5. As the study relies heavily on Dex-inducible constructs and artificial miRNAs, the authors should include controls to address potential off-target or Dex-related effects. For example, assessing embryo sac development in lines expressing unrelated amiRNAs or Dex-inducible reporters not targeting H1.1 would help establish the specificity of the observed phenotypes.

>> We addressed this question by assessing embryo sac development following Dex induction of the wild-type H1.1-RFP variant, which does not cause any defects in embryo sac development, **as shown in Schubert et al, Plant Methods 2022 (Schubert et al, 2022), and in Figure 5**. The scoring of FG2/FG4 stages versus mature FG6 stage is similar in Dex induced controls and in mock-induced flowers (**Figure EV5C**).

Referee #2:

This study investigates the role of linker histone H1 eviction during differentiation of spore mother cells (SMCs) in *Arabidopsis thaliana*. The authors propose that H1 eviction is controlled by two post-translational modifications: (i) ubiquitination at K89, mediated by

the CULLIN4 E3 ligase, which targets H1.1 for degradation via the proteasome; and (ii) citrullination at R57, which destabilizes H1.1 by weakening its binding to chromatin. They propose that a potential plant citrullinase, AIH, catalyzes citrullination at R57. Downregulation or inhibition of AIH (by Cl-amidine) partially impairs H1.1 depletion. Furthermore, the authors find that H1.1 depletion is crucial for gametogenesis but not for meiosis. Based on these findings, they propose a two-step eviction model: in step 1, citrullination destabilizes H1.1 and increases the unbound pool of H1; in step 2, the unbound H1.1 is ubiquitinated by CUL4 and subsequently degraded. While the research question is relevant and interesting, I have several concerns regarding the data and the conclusions that would need to be addressed.

1. AIH and CUL4 are broadly expressed, yet H1.1 eviction is specific to the SMC. What confers this specificity?

>> We kindly refer Reviewer 2 to our response to Reviewer 1, which addresses the same question (Point 2).

2. The authors propose that a single H1 arginine residue is the primary target for citrullination. However, mutating R57 to K results in only 26% of SMCs with persistent H1.1, a relatively minor effect compared to the control (11% persistent). If citrullination of H1.1 precedes CUL4-mediated degradation, one would expect that blocking citrullination would have a similar consequence to blocking CUL4-yet this is not the case.

>> We attribute the moderate (26% compared to 3% and 7% in the controls and not 11%, **Figures 2B** and **EV2C**) but significant persistence rate of H1.1-R57K to a model in which the R57K mutation does not block degradation per se but rather slows the process by reducing the rate at which H1.1 is released from chromatin and becomes accessible to the degradation machinery. We have **revised the text** on page 20 of the discussion to **clarify our model**: “In contrast, neutralizing the charge by an alanine substitution (R57A) creates an H1.1 variant with a higher mobility, i.e. a shorter residency time on the chromatin as the wild-type H1.1. While the R57K substitution protects H1.1 from degradation in a significant fraction of SMC, the R57A substitution does not. In contrast, the latter seems to induce a faster depletion as suggested by the reduced fraction of SMC with partial depletion at the time of scoring (Figure EV2A). This finding indicates that H1.1 binding properties play a role in its degradation. We thus propose a model (Figure 6) in which charge neutralization at R57 increases the dissociation rate of H1.1 from chromatin, making it more susceptible to ubiquitination-mediated degradation. This neutralization, for example, occurs through citrullination in wild-type SMCs, leading to rapid H1.1 depletion in a normal context. Similarly, the R to A substitution in the H1.1R57A mutant results in increased mobility, facilitating degradation. In contrast, retaining a positive charge at R57, as in cells where H1.1 remains unmodified or following the R to K substitution in the H1.1R57K variant, slows dissociation, resulting in a higher proportion of SMCs with detectable H1.1 levels at a given time. However, this substitution does not stabilize H1.1 on chromatin. The H1.1R57K variant exhibits similar mobility to the wild-type protein but dissociates more slowly than the citrullinated variant, eventually leading to degradation. This explains why we do not observe persistent H1.1 in

all SMCs, by contrast to SMCs expressing H1.16GC or H1.1K89R mutant variants that are protected from degradation.”

It would therefore be important to also test the effect of the R79K mutation, which, as the authors note, is the corresponding citrullinated residue in mouse H1.

Since submission, we have also tested the effect of the R79K mutation and observed no impact on H1.1 stability in the SMC, based on analysis of over 260 ovule primordia from six independent lines. These **new results have been included in Figure EV2B**, and a corresponding description has been **added to the main text** on page 10 “To investigate this, we designed an H1.1 mutant with an R-to-K substitution preserving the positive charge, at the residue R79 located inside the globular domain, and which is conserved with the mouse homolog R52 residue (Figure EV2A). We co-expressed H1.1R79K-RFP under the control of the pOP/LhGR Dex-inducible system and H1.1-GFP as a control (native H1.1 variant expressed under its own promoter, She *et al*, 2013). H1.1R79K-RFP showed depletion in the SMC as did H1.1-GFP (n=260 ovule primordia, 6 independent lines, Figure EV2B). Then we mutated another conserved arginine located just before the globular domain, at position R57. This time, the R57K mutation compromised H1.1 depletion in a large fraction of ovule primordia (26%, n=218 Figure 2B and 26%, n=209 Figure EV2A)’.

The statement "the R57K mutation compromised H1.1 depletion in a large fraction of ovule primordia" should be toned down and adjusted to the findings.

We have now **adjusted the sentence** for “the R57K mutation compromised H1.1 depletion in a significant fraction of ovule primordia (26%, n=218 Figure 2B and 26%, n=209 Figure EV2C) compared to the control (7% and 3%, Figure 2B and EV2C)”

3. Figure 1C: The 11xG mutants are not described in the Results section. It is difficult to understand how the K89R mutation has an effect, while mutation of 11 lysine residues, including K89, does not. These data are mentioned in the discussion, but should be addressed in the Results section as well.

>> Like the reviewer, we were also surprised by this finding. To address this, we sequenced the transgene from all 12 independent transgenic lines that were screened and confirmed they contained the 11 lysine substitutions. We **provide a new panel in figure EV2K** showing this result as well as representative images of H1.1-11G-RFP in ovule primordia showing generally lower abundance, possibly suggesting a lower viability of this variant. We have now **clarified** this point in the **Results and discussion.** Intriguingly, the H1.111xG variant, which also contains the K89R substitution, did not exhibit the persistence phenotype observed with H1.1K89R. Sequencing of the transgenic insertions confirmed the presence of the mutations in 11 lines (Figure EV1K). The generally lower signals obtained for this mutant variant (Figure EV1K) suggest lower viability due to cumulated mutations, which may mask the specific effect of K89R.” and “Intriguingly, a mutated H1.1 variant (H1.111xG, Figure 1) with 11 substituted lysines residues in the globular domain, confirmed by sequencing the transgenes, and including the K89R mutation, exhibited normal depletion in the SMC. While this suggests that K89’s role in degradation requires an intact globular domain, it is unlikely that the 11 K-to-R

substitutions disrupt its folding, as this domain is highly resilient to amino acid changes (Martinsen *et al*, 2022). The apparent lower abundance of H1.111xG levels compared to other variants may suggest a lower viability, stability or both, of this mutant variant masking the specific effect of K89R. At present we cannot fully explain this observation but it suggests that H1.1 degradation may be subject to versatile and redundant mechanisms involving concurrent or cooperative PTMs. In support of this hypothesis, H1.1 in leaves was found to carry a wide variety of PTMs, including crotonylation at six lysines.(Kotliński *et al*, 2016).”

4. Figure 5C: The R57K line exhibits a high level of seed abortion, which appears inconsistent with its relatively minor effect on female gametophyte development. Furthermore, to show that the 6xGC line indeed impairs female gametophyte function, the authors should pollinate this line with wild-type pollen and then score seed development.

>> As discussed on page 17, the high rate of seed abortion in the R57K line may be attributed to the low viability of pollen, as shown in **Supplementary Figure S5**. An effect at the time of pollination itself cannot be ruled out, but it would require further investigation beyond the scope of this study. The suggestion to pollinate the 6xGC line with wild-type pollen would be relevant only if this line produced a mature gametophyte containing both the egg and central cells, which is not the case. Therefore, we believe this experiment would not provide additional insights.

5. Based on the claims of the authors, mutations in CUL4 should lead to a female gametophytic effect. However, this was not tested by the authors; instead they tested seed abortion upon self-pollination, which does not allow to conclude whether the female gametophyte is functionally impaired. They should pollinate their lines with wild-type pollen to test whether this impairs seed development. It would add to this manuscript if the authors would include the hypomorphic *cul4-1* allele (Bernhardt *et al.*, 2006), which should recapitulate the findings reported using the amiR-lines.

>> Reproductive defects associated with CUL4 depletion or reduced expression have been reported in several studies. One study (Chen *et al*, 2006) described shorter siliques, while another (Dumbliauskas *et al*, 2011) reported both zygotic embryo lethality and gametophytic lethality, as evidenced by reduced segregation on both the male and female sides following reciprocal crosses of *cul4-2* and *cul4-3* mutants with wild-type ovules or pollen, respectively. This same study also reported in their Supplementary Figure S1 the occurrence of infertility in ovules for all three *cul4* mutant alleles (*cul4-1*, a hypomorphic mutant, and *cul4-2* and *cul4-3*, both loss-of-function mutants). Had we used the hypomorphic mutant directly, we would likely have been critiqued for the lack of specificity regarding the developmental stage we are analyzing, potentially masking stage-specific effects. Moreover, we would likely have been asked to develop an alternative strategy providing better temporal and spatial control over CUL4 downregulation. We trust that the reviewers will now also appreciate that the suggested experiment would not provide additional insights to the current study and that our inducible amiRNA-based approach to downregulate CUL4 specifically in developing ovule primordia remains the most optimal and focused method for this investigation.

We now clarified it in the text page 5: “CUL4 loss-of-function and hypomorph mutations show reproductive defects with both zygotic and gametophytic lethality (Chen *et al*, 2006; Dumbliauskas *et al*, 2011). To avoid confounding effects, we conditionally knocked down CUL4 expression in developing ovule primordia just prior to the onset of H1 eviction.”

6. Please correct: Female gametophyte development does not involve endomitosis (DNA replication without subsequent mitosis), and the female gametophyte before cellularization is a coenocyte, not a syncytium (which is a structure developing upon cell fusion).

>> we corrected the terminology as suggested.

Referee #3

This work is aimed at understanding the loss of histone H1 in SMC maturation, a process already reported earlier by the same laboratory. Now authors develop an elegant strategy to establish that H1 is degraded via the proteasome which is analogous to the H1 modification and degradation reported in the mouse. They found that this is dependent on CUL4 and identify two amino acid residues necessary for efficient degradation, K79 and R57. In addition to ubiquitination of K79 that marks H1 for proteolysis, citrullination of R57 is necessary for H1.1 depletion. This step is carried out by AIH, an arginine deiminase. They also try to delineate the functional relevance of these modifications and found that H1 persistence does not impair SMC maturation or meiosis but are important for SMC growth and later establishment of the embryo sac and seed fertility.

The work is experimentally correct and I do not have major concerns. However, addressing my comments below may improve the quality and impact of the study.

1. Section describing the screen for potential Ubiquitination sites. The rationale to select the residues to be mutated seems to be based on homology studies. Is this the case?

We **amended the explanation in the text** with more details about the residues chosen based on different predictive approaches “Candidate lysine were selected based on computational predictions (AtH1.1 K134, 139, 144, 172, 177, 179, 185, 189, 191, 193, 204, 211, 213, 215, 223, 226, 232, 273, 274, Chen *et al*, 2011; Radivojac *et al*, 2010; Walton *et al*, 2016), proteomic evidence of ubiquitination in seedling tissues (AtH1.1 K89, 204, 206, 211, 213, 215, **Figure EV1E**, (Walton *et al*, 2016) and comparison with ubiquitinated sites in mouse and human counterparts (AtH1.1 K77, 89, 104, 111, Wiśniewski *et al*, 2007). We also included in this list of candidate residues, several lysins in the N-terminal tail upstream the globular domain to cover all three functional domains of the protein in our targeted mutagenesis approach. Several mutant H1.1 variants were thus designed...”

It could be worth explaining in more detail the process to select a single amino acid residue, e.g., K79, for further analysis.

>> we are not sure if the reviewer refers to K89 or R79 – which was initially not tested. How we identified K89 as an important residue came from the analysis of two variants carrying

either one mutation in the globular domain (K89R) or five in the C-terminal tail (5xC), as **explained already** in the text “To further distinguish the contribution of the different lysine mutated in the H1.16xGC variant, we created two additional mutants: one carrying a single mutation at lysine 89 (K89R) in the globular domain (H1.1K89R, Figure 1C, Figure EV1H) and another with K-to-R mutations at the five remaining lysines in the C-terminal tail (H1.15xC, Figure 1C).”

2. Impairment of H1 depletion only moderately affects SMC growth but not expression of specific markers. Given the importance of chromatin remodeling this seems unexpected and may deserve a more detailed discussion.

>> we thank the reviewer for the opportunity to clarify the discussion. We **revised the last section of the discussion page 24** and hope to have met the reviewer’s expectations.

“Notably, depletion of linker histones at the onset of SMC differentiation precedes the gradual decrease in constitutive heterochromatin (chromocenters) and in H3K27me3 (She *et al*, 2013) as well as a decrease in mCHH (Ingouff *et al*, 2017), compared to surrounding cells. SMC expressing H1.1R57K show moderately - but significantly-elevated levels of these chromatin features as measured by the relative heterochromatin fraction and by fluorescently tagged H3K27me3 and mCHH readers compared to wild-type. By contrast, the expression of H1.16GC led to little or no effects on chromatin reorganisation. These distinct outcomes can be explained through our model (Figure 6) as follows: the slower dissociation rate of H1.1R57K mutant, compared to the citrullinated wild-type H1.1 variant (H1.1cit), leads to prolonged chromatin residency. This extended binding time may partially hinder or delays heterochromatin decondensation, resulting in increased RHF levels as measured. Similarly, it may partially hinder the global reduction in H3K27me3 and mCHH which in turn leads to elevated levels of their respective reporters. By contrast, the H1.16GC variant retains the ability to be citrullinated at R57, resulting in a higher dissociation rate and shorter chromatin residency time -similar to H1.1cit- which may be favorable to chromatin reorganisation. These findings highlight the role of H1.1 removal—initiated by its increased dissociation rate and culminating in degradation—in promoting chromatin reorganization in SMCs, a process that likely creates a window of opportunity for epigenetic reprogramming during the somatic-to-reproductive fate transition.”

3. H1 depletion appears to be important for the embryo sac development. Since this step requires multiple divisions of the haploid nuclei and H1 contains several putative CDK phosphorylation sites it could be relevant exploring a possible relationship with H1 modifications. I understand that this is beyond the scope of the present study but I wonder if roscovitine could be used to answer this.

>> The reviewer is correct in noting that H1.1 contains several phosphorylation sites, and phosphorylation is indeed a modification that destabilizes H1 in animal cells during replication. In plants, this mechanism has not been addressed yet. Using a general CDK inhibitor or CDK mutants would likely introduce a range of indirect effects on meiosis as shown (Yang *et al*, 2020) and gametophyte development, making it difficult to isolate the specific role of H1. We therefore suggest that future investigations focus on mutating the SP/xT phosphorylation sites directly. While such an approach would be an interesting and

potentially complementary study to our current model—though it may not yield the expected results—it falls outside the scope of this study.

Page 22, we already discuss “For instance, phosphorylation is a major destabilizing PTM of H1 at the onset of S-phase and crucial for replication in animal cells (Alexandrow & Hamlin, 2005). The Arabidopsis H1.1 variant contains three S/TPxK motifs, which are prone to phosphorylation in the C-terminal tail (Kotliński et al, 2016), and CDC2/CDKA;1 can physically interact with H1.1 (Pusch, 2012). It remains to be determined whether this mechanism contributes to H1.1 depletion in the SMC prior to meiotic S-phase.”

4. It may be worth including in the discussion a recent article (• DOI: 10.1038/s41586-025-08835-0) on H1 deamidation and its role in chromatin relaxation.

>> We thank the reviewer for bringing this article to our attention, which we have read carefully. We would like to clarify that H1 deamidation, which converts asparagine into aspartic acid, is distinct from deimination, which converts arginine into citrulline. Additionally, deamidation appears to act as a substrate for downstream acetylation, promoting chromatin relaxation, whereas in our model, citrullination facilitates exposure to the ubiquitin ligase CUL4 for degradation. Furthermore, while H1 deamidation in the referenced study is associated with transient and localized chromatin relaxation during DNA repair, our study focuses on H1 citrullination, which promotes degradation and drives global chromatin changes. Although we found this study very interesting, we did not identify a clear angle to incorporate it into our discussion. Nevertheless, we appreciate the reviewer highlighting this valuable reference for potential future work.

However, we have taken the opportunity to **revise the discussion** to explore potential mechanisms through which H1 removal could impact chromatin organization and gene expression:

Page 25 “In Arabidopsis, H1 variants are not deposited at specific loci but are instead broadly distributed across the genome, with a notable enrichment at lowly expressed genes (Rutowicz et al, 2015). As such, the impact of H1 removal in the SMC on genes involved in embryo sac formation and gametogenesis remain unclear. In mammals, H1 variants are similarly distributed across the genome, and their reduced stoichiometry results in gene expression changes associated with local chromatin decompaction, with levels of H3K27me3 and H3K36me3 decreasing and increasing, respectively (Willcockson et al, 2021). In Arabidopsis, H1 depletion moderately affects gene expression and chromatin accessibility, particularly at loci targeted by the Polycomb Repressive Complex 2 (PRC2)(Rutowicz et al, 2019; Teano et al, 2023). It also broadly influences accessibility to the chromatin remodeler DDM1, which plays a key role in DNA methylation (Zemach et al, 2013). Determining whether H1.1 removal in the SMC influences gene expression globally or at specific loci by modulating chromatin accessibility to chromatin modifiers like DDM1 and PRC2 will be an intriguing, yet challenging, avenue for future investigation. Alternatively, and not exclusively, a model in which H1.1 competes with transcription factors for DNA binding could offer a mechanism to control a subset of embryo sac-specific genes. This hypothesis is supported by the observation that H1 depletion primarily impacts NAC target genes among diurnally regulated genes (Rutowicz et al, 2025). In apple, a linker histone variant was also shown to act as a direct transcriptional regulator of metabolic genes (Hu et al, 2025). Thus,

exploring whether H1.1 depletion affects a specific set of loci enriched with DNA-binding motifs, through cell-specific transcriptome profiling in the embryo sac, would provide valuable insights.”

5. The mechanism proposed here is quite sophisticated. Do the authors consider that it is applicable exclusively to SMC or may operate also at other stages of plant life or other organs? It is known that a different H1 variant (H1.3) is expressed in response to abiotic stress and likely replaces H1.1 and H1.2. Could a similar mechanism apply in seedlings in response to abiotic stress? I suggest including this in the discussion.

>> We thank the reviewer for this suggestion. We **added a concluding remark** at the end of the discussion “Finally, future investigations should explore whether H1 citrullination and ubiquitination also influences the binding dynamics, abundance and stoichiometry of all three H1.1, H1.2 and H1.3. Several phenotypes resulting from H1 depletion have been described in Arabidopsis, such as altered development (flowering time, lateral root and root hair abundance, stomatal patterning, Rutowicz et al. 2019) and environmental responses (to drought and low light, salt stress, heat stress, and defense priming, (Rutowicz et al, 2015; Liu et al, 2021; Sheikh et al, 2023; Perrella et al, 2024). Yet, the specific role of H1 variant stoichiometry—i.e., the delicate balance in abundance and genomic distribution of all three variants in a cell and tissue-type-specific manner during development and biotic and abiotic stress responses —has yet to be explored. “

6. There are a few typos. Page 9, line 24, amin... ; Page 16, line 33, Too ... Please check throughout.

>> thank you for notifying these typographic mistakes which are now corrected.

Dear Dr. Baroux,

Thank you for submitting a revised version of your manuscript. Your study has now been seen by all original referees. Referees #1 and #3 find that their previous concerns have been addressed and now recommend publication of the manuscript. Referee #2 still voices three concerns which I would ask you to address in the final version of the manuscript. The concerns regarding the response to comment #1 and #5 ask for additional discussions but the concern regarding your response to comment #4 would require additional experiments. We agree with this referee that the data would benefit the manuscript should such an experiment have been started or the reciprocal crosses being available already. We would not insist here if this experiment would have to be started freshly.

In addition, there remain only a few mainly editorial points that have to be addressed before I can extend formal acceptance of the manuscript:

- Thank you for providing your data. Please make sure that figure panels are clearly labeled in zip folder, not only in Excel files.
- Thank you for uploading high resolution EV figures. Please add subtitle for EV figure legends in the manuscript file.
- Please double-check to make sure that all relevant funding information in the manuscript is also entered into our submission system. (Missing in the system currently: grants #310030_185186 and #31003A_179553 for the University of Zurich, the Swiss National Science Foundation; the Marie Skłodowska-Curie (MSC) grant agreement No 847585)
- Please reduce the number of keywords on the abstract page to five (ideally choosing broad general terms) and place these below the Abstract
- As we are switching from a free-text author contribution statement towards a more formal statement based on Contributor Role Taxonomy (CRediT) terms, please remove the present Author Contribution section and instead specify each author's contribution(s) directly in the Author Information page of our submission system during upload of the final manuscript. See <https://casrai.org/credit/> for more information.
- Please adjust the in-text callouts for individual figures and figure panels: e.g. it is unclear what "Supplemental data 1" is referring to and there is also a missing callout for Fig. 2a; in Fig. 2 legend panel C is missing
- Please provide suggestions for a short 'blurb' text prefacing and summing up the conceptual aspect of the study in two sentences (max. 250 characters), followed by 3-5 one-sentence 'bullet points' with brief factual statements of key results of the paper; they will form the basis of an editor-written 'Synopsis' accompanying the online version of the article. Please also provide an altered synopsis image, making sure that the aspect ratio conforms to our website's format - it should be exactly 550 pixels wide and between 300-600 pixels high.
- We have noticed that there is reuse of figures between primary and EV figures. (Figure 2B and Figure EV2B - 3C,D and Figure EV3D,E - Figure 4D and Figure EV4C - Figure 5A and Figure EV5A and finally within Figure EV3B). Please detail the reuse in the corresponding figure legend.
- Please note that the specific URL for S-BIAD2120 dataset is not provided in the data availability statement.
- Please correct the Sections order to: Title page - Abstract - Keywords - Introduction - Results - Discussion - Methods - Data Availability - Acknowledgements - Disclosure and Competing Interests Statement - References - Figure Legends - Table(s) - Expanded View Figure Legends.
- Figure Legends (main + EV):
 1. Please note that the figure 2C is mislabeled as figure 2B in the manuscript. This needs to be rectified.
 2. Please note that the exact p values are not provided in the legends of figures 1D, 2B, 3C, D; 4A-C, G; EV2 E
 3. Please indicate the statistical test used for data analysis in the legend of figure EV1 C
 4. Please note that the box plots need to be defined in terms of minima, maxima, centre, bounds of box and whiskers, and percentile in the legends of figures 4A-C, G; 5C, EV1 C, F; EV2 E, EV4 A-C.
 5. Please note that information related to n is missing in the legends of figures EV1 F, EV2 E.

6. Please note that the error bars are not defined in the legends of figures 1E, 2C, EV1 G.
7. Please note that the scale bar needs to be defined for figures 4A-C
8. Please note that scale bar and its definition are missing for figures EV5 A, B
9. Please note that the white arrows are not defined in the legend of figure EV1 D.
10. Please note that the dotted borders are not defined in the legend of figures 1A, B, D; 2B, 3C, D; 4A-C; 5B; EV1 I, K; EV2A, D.

With best regards,
Cornelius Schneider

Cornelius Schneider, PhD
Editor | The EMBO Journal
c.schneider@embojournal.org

Please refer to our figure preparation guideline in order to ensure proper formatting and readability in print as well as on screen:

See also figure legend guidelines:

<https://www.embopress.org/page/journal/14602075/authorguide#figureformat>

Use the link below to submit your revision:

Referee #1:

The authors have adequately addressed the concerns raised in the previous round of review.

Referee #2:

Response to comment 1:

The authors discuss several hypothetical possibilities that may account for the increased turnover of H1 in the SMC. Is there any supportive evidence for these scenarios (Ca²⁺ availability or ow or stalled transcription and/or translation)?

Response to comment 4:

Based on the data presented in Supplemental Figure 5 and the accompanying text, the R57K line exhibits only a minor fraction of aborted pollen, which does not sufficiently explain the observed high level of seed abortion (contrary to the authors' response). It would therefore be informative to assess whether the pollen is indeed functional by performing reciprocal crosses between R57K and wild-type plants. I leave it up to the authors to consider this straightforward experiment to clarify the issue and strengthen the manuscript.

Response to comment 5:

Published data show that loss-of-function alleles cul-3 and cul-4 exhibit only a minor, non-significant defect in maternal transmission. This observation contrasts with the authors' claim that loss of CUL4 impairs female gametophyte function. It is therefore important that the authors clarify this point and accurately reflect the published findings in the manuscript. The available data instead suggest a predominantly sporophytic role of CUL4, which should be discussed accordingly.

Referee #3:

Authors have addressed adequately my comments. They have modified, clarified and expanded the text, as suggested.

Manuscript text revisions – Reviewer #2**Reviewer Response to comment 1:**

The authors discuss several hypothetical possibilities that may account for the increased turnover of H1 in the SMC. Is there any supportive evidence for these scenarios (Ca²⁺ availability or low or stalled transcription and/or translation)?

Author response:

Page 17, in the discussion, we reviewed already all known observations on these topics:

- regarding transcription, we have previously shown that the levels of active RNA Pol II is lower in the female SMC compared to surrounding cells (She et al. 2013). And we wrote “However, the short lifetime of H1.1 transcripts in the SMC progenitor, in combination with a relative transcriptional quiescence (She et al, 2013) may be sufficient to prevent the replenishment of H1.1 on chromatin.” We now added “of the SMC” before (She et al.) to be clearer

-regarding translation. We discussed the following “Whether translational inhibition also occurs in the SMC, targeting specific transcripts such as those encoding H1.1, or if it is part of a broader mechanism—similar to what is seen in animal PGCs (Oulhen et al, 2017)—remains to be determined. Interestingly, premeiotic spikelet and male meiocytes in prophase I of rice are enriched in phased secondary small interfering RNAs (phasiRNAs) involving MEL1 (Komiya et al, 2014) and bearing the potential for translational inhibition (Jiang et al, 2020; Liu et al, 2020). Future studies could explore the repertoire of small and antisense RNAs in the female SMCs of Arabidopsis». We now added « that could influence transcription and translation» at the end of the last sentence to be clearer.

-regarding Ca²⁺ availability, we already indicate what we found : “Although calcium showed undetectable levels in male SMCs in *Nicotiana tabacum* and *Torenia fournieri* (Ge et al, 2008), no studies have yet been conducted on calcium availability in ovule primordia or in SMCs.”

Reviewer Response to comment 4:

Based on the data presented in Supplemental Figure 5 and the accompanying text, the R57K line exhibits only a minor fraction of aborted pollen, which does not sufficiently explain the observed high level of seed abortion (contrary to the authors' response). It would therefore be informative to assess whether the pollen is indeed functional by performing reciprocal crosses between R57K and wild-type plants. I leave it up to the authors to consider this straightforward experiment to clarify the issue and strengthen the manuscript.

Author response:

We indeed wrote page 12 “We observed 12% aborted pollen (n=323) in anthers expressing H1.1R57K-RFP at 9dpi compared to 3% in the control line (n=425, Figure EV5D). This pollen abortion rate cannot explain the high frequency of aborted seeds. Hence, whether the induction of the H1.1R57K variant affects gametic functionality or that of sporophytic tissues remains to be investigated.”

As this study is centered on the mechanisms and roles of H1 depletion in female SMCs, we limited our analyses of male tissues to relevant supporting observations. A detailed investigation of H1 dynamics in pollen viability and function falls beyond the intended scope of this work and will be pursued in future research

Reviewer Response to comment 5:

Published data show that loss-of-function alleles cul-3 and cul-4 exhibit only a minor, non-significant defect in maternal transmission. This observation contrasts with the authors' claim that loss of CUL4 impairs female gametophyte function. It is therefore important that the authors clarify this point and accurately reflect the published findings in the manuscript. The available data instead suggest a predominantly sporophytic role of CUL4, which should be discussed accordingly.

Author response:

We believe that the reviewer refer to the sentence page 5, lines 20-22 “CUL4 loss-of-function and hypomorph mutations show reproductive defects with both zygotic and gametophytic lethality (Chen et al, 2006; Dumbliauskas et al, 2011).”

The genetic analysis of the loss-of-function alleles *cul4-2* and *cul4-3* (Dumbliauskas et al., 2011, Table 1) reported transmission efficiencies of 84% and 87% through the female gametophyte and 85% and 90% through the male, respectively. These values correspond to gametophytic lethality of approximately 13–16% for ovules and 10–15% for pollen. The authors concluded that “the transmission efficiency of the marker was slightly reduced through both male and female gametophytes.” We note, however, that the reported P values (0.06–0.1 for female TE and 0.077–0.1 for male TE) indicate that the reductions are not statistically significant.

Nonetheless, to acknowledge this trend, we have revised our wording to “mild gametophytic lethality (10–15%).” Furthermore, as the same study reported seed abortion in this line—indicative of zygotic-effect embryo lethality—we have clarified our description accordingly, replacing “zygotic lethality” with “embryonic lethality.” Finally, we have now included mention of the sporophytic roles as suggested.

We now modified the sentence for “CUL4 is critical to plant development and its downregulation leads to pleiotropic defects in leaves and roots as well as severe sterility (Chen et al., 2006; Bernhardt et al., 2006). Complete loss of function is embryo-lethal and is also associated with a mild reduction (10–15%) in transmission efficiency through both male and female gametes, indicating a low level of gametophytic lethality (Dumbliauskas et al., 2011).”

Dear Dr. Baroux,

I am pleased to inform you that your manuscript has been accepted for publication in the EMBO Journal.

Yours sincerely,

Cornelius Schneider, PhD
Editor
The EMBO Journal
c.schneider@embojournal.org

Please note that it is The EMBO Journal policy for the transcript of the editorial process (containing referee reports and your response letters) to be published as an online supplement to each paper. If you should prefer removal of any referee-only figures included in the point-by-point response(s), e.g. because they may still be used for future publication or because they have been reproduced from published work by others, please do let us know immediately via response email.

More information is available here: https://www.embopress.org/transparent-process#Review_Process
